# Systematic genetics and single-cell imaging reveal widespread morphological pleiotropy and cell-to-cell variability

Mojca Mattiazzi Usaj[1,†], Nil Sahin[1,2,†], Helena Friesen[1], Carles Pons[3], Matej Usaj[1], Myra Paz D Masinas[1], Ermira Shuteriqi[1], Aleksei Shkurin[1,2], Patrick Aloy[3,4], Quaid Morris[1,2,5,*], Charles Boone[1,2,6,**] & Brenda J Andrews[1,2,***]

## Abstract

Our ability to understand the genotype-to-phenotype relationship is hindered by the lack of detailed understanding of phenotypes at a single-cell level. To systematically assess cell-to-cell phenotypic variability, we combined automated yeast genetics, high-content screening and neural network-based image analysis of single cells, focussing on genes that influence the architecture of four subcellular compartments of the endocytic pathway as a model system. Our unbiased assessment of the morphology of these compartments—endocytic patch, actin patch, late endosome and vacuole—identified 17 distinct mutant phenotypes associated with ~1,600 genes (~30% of all yeast genes). Approximately half of these mutants exhibited multiple phenotypes, highlighting the extent of morphological pleiotropy. Quantitative analysis also revealed that incomplete penetrance was prevalent, with the majority of mutants exhibiting substantial variability in phenotype at the single-cell level. Our single-cell analysis enabled exploration of factors that contribute to incomplete penetrance and cellular heterogeneity, including replicative age, organelle inheritance and response to stress.

**Keywords** cell-to-cell variability; endocytosis; high-content screening; phenotype classification; single-cell analysis
**Subject Categories** Computational Biology; Organelles
**Mol Syst Biol. (2020) 16: e9243**

## Introduction

Although we understand that most phenotypes including diseases are influenced by the genetic variation encoded in individual genomes, our ability to predict when a genetic lesion will cause a specific phenotype remains limited. Pioneering work in yeast and other model systems has made use of quantifiable phenotypes, such as cell growth, to systematically survey the consequences of single, double and higher-order genetic perturbations in populations of mutant cells (Costanzo *et al*, 2019; Domingo *et al*, 2019). While measuring growth phenotypes in cell populations has enabled inference of gene function, biological pathways and networks, the mechanistic underpinnings of a particular phenotype are typically difficult to infer from bulk population measurements. Moreover, using population-level measurements as a phenotypic read-out precludes a quantitative analysis of single-cell phenotypes and thus an analysis of cell-to-cell variability, which is a key consideration for prediction of the consequences of genetic perturbation.

High-throughput (HTP) approaches for monitoring single-cell phenotypes include single-cell transcriptomics, mass spectrometry and automated imaging, among others (Ziegenhain *et al*, 2018; Chessel & Carazo Salas, 2019; Yin *et al*, 2019). High-content screening, which combines HTP microscopy with multiparametric image and data analyses, provides rich phenotypic information about the spatio-temporal properties of biological systems at the single-cell level (Boutros *et al*, 2015; Mattiazzi Usaj *et al*, 2016; Chessel & Carazo Salas, 2019). Large-scale screens have been productively combined with image analysis to explore different aspects of cell biology in yeast and in higher eukaryotes. For example, data on protein localization and abundance, cell shape and compartment morphology, and the prevalence of cell-to-cell variability can be

---

1 The Donnelly Centre, University of Toronto, Toronto, ON, Canada
2 Department of Molecular Genetics, University of Toronto, Toronto, ON, Canada
3 Institute for Research in Biomedicine (IRB Barcelona), The Barcelona Institute for Science and Technology, Barcelona, Catalonia, Spain
4 Institució Catalana de Recerca i Estudis Avançats (ICREA), Barcelona, Catalonia, Spain
5 Computational and Systems Biology Program, Memorial Sloan Kettering Cancer Center, New York, NY, USA
6 RIKEN Centre for Sustainable Resource Science, Wako, Saitama, Japan
  *Corresponding author. Tel: +1 416 978 8568; E-mail: morrisq@mskcc.org
  **Corresponding author. Tel: +1 416 978 6113; E-mail: charlie.boone@utoronto.ca
  ***Corresponding author. Tel: +1 416 978 8562; E-mail: brenda.andrews@utoronto.ca
  †These authors contributed equally to this work

quantified from cell images and the influence of genetic or environment perturbation on these cell attributes can be systematically assessed (Yin *et al*, 2013; Chong *et al*, 2015; Styles *et al*, 2016; de Groot *et al*, 2018; Heigwer *et al*, 2018).

Endocytosis is a highly conserved bioprocess that plays a central role in eukaryotic cell biology, mediating the internalization of receptors, nutrients and other molecules, controlling the lipid and protein composition of the plasma membrane and the coupling of different intracellular compartments (Goode *et al*, 2015). Endocytosis initiates with vesicle formation at specific sites at the plasma membrane. This process requires the coordinated action of proteins involved in distinct functional modules (Lu *et al*, 2016). In yeast, these include coat proteins, which function as adaptors to link cargo, coat, plasma membrane and actin network components, and actin module proteins, which represent a later stage in internalization; their appearance coincides with the membrane invagination and coat internalization step (Weinberg & Drubin, 2012). After cargo uptake, endocytic vesicles fuse with early endosomes, allowing cargo to be recycled to the plasma membrane, or targeted through more mature endosomes and multivesicular bodies (MVBs) for vacuolar (lysosomal) degradation. The endocytic intracellular trafficking pathway, which is largely recapitulated in mammalian cells (Taylor *et al*, 2011), impinges on a number of cellular physiological processes and is often associated with the pathology of human diseases, including atherosclerosis, some cancers and Alzheimer's disease (McMahon & Boucrot, 2011; Maxfield, 2014). Several large-scale studies have been conducted to identify a number of core components and regulators of the endocytic pathway in yeast and higher eukaryotes, but have largely been based on population measurements or have analysed only a subset of genes (Bonangelino *et al*, 2002; Seeley *et al*, 2002; Burston *et al*, 2009; Collinet *et al*, 2010; Liberali *et al*, 2014).

To explore how single-cell analysis can be used to assess cell-to-cell variability, morphological pleiotropy and incomplete penetrance, we used the yeast endocytic pathway as a model system, combining systematic genetic analysis with high-content screening. We examined 5,292 unique yeast genes for roles in endocytic compartment morphology, applying live-cell fluorescence microscopy and neural network-based, single-cell image analysis. In total, we identified ~1,600 genes whose perturbation affects at least one endocytic compartment, revealing both new biology and insights into mechanisms underlying cellular heterogeneity. The experimental and computational pipeline developed here can be generalized to other unrelated compartments, pathways or phenotypes, which will allow us to expand our knowledge on the inner workings of a cell. Importantly, the computational analysis framework we developed is also species-independent, and we provide the tools for its implementation.

# Results

### Combined experimental–computational pipeline for quantitative single-cell assessment of mutant phenotypes

To enable a quantitative analysis of subcellular compartment morphology, we developed a high-throughput (HTP) image-based pipeline coupled to single-cell image analysis (Fig 1A). We constructed a series of query strains with a fluorescent protein (FP)

at the C terminus of four endogenous yeast proteins, each serving as a marker for a unique endocytic compartment. We focused on (i) *SLA1*, encoding an endocytic adaptor protein, marking the coat complex associated with early endocytic sites at the plasma membrane; (ii) *SAC6*, encoding yeast fimbrin, marking actin patches that are also required for early endocytosis events; (iii) a late endosomal marker, *SNF7*, encoding a subunit of the ESCRT-III complex involved in the sorting of transmembrane proteins into the multivesicular body (MVB) pathway; and (iv) a marker for the vacuolar membrane, *VPH1*, encoding subunit "a" of the vacuolar ATPase (V-ATPase) $V_O$ domain (Fig 1B).

We introduced each marker into both the yeast deletion collection (Giaever *et al*, 2002), and the collection of temperature-sensitive (TS) mutants of essential genes (Li *et al*, 2011; Costanzo *et al*, 2016), using the synthetic genetic array (SGA) approach (Tong & Boone, 2006). We acquired live-cell images of log-phase cultures with an automated HTP microscope. CellProfiler (Carpenter *et al*, 2006) was used to identify individual cells and subcellular compartments, and extract quantitative features describing these segmented compartments. The final dataset included quantitative data for ~16.3 million cells from 5,627 mutant strains (5,292 unique ORFs or ~90% of yeast genes), with an average of 640 cells for each mutant strain.

First, we defined the phenotypes associated with each compartment. We used an automated unsupervised method to identify "outlier" cells with non-wild-type morphology (see Materials and Methods). To identify mutant morphologies, we visually inspected the strains with a significant fraction of outlier cells, assessed their phenotypes and compiled a set of positive control strains by combining published data with selected mutants (Table EV1). This approach enabled the discovery of both well-characterized and novel phenotypes. In total, we defined 21 endocytic phenotypes: a wild-type phenotype for each compartment and 17 showing aberrant morphology (Fig 1C).

We then labelled a representative set of cells displaying these 21 phenotypes using a custom-made, single-cell labelling tool; this "training set" was used to train a neural network to automatically classify other cells. To confirm that the CellProfiler features derived from the cell images were sufficient to distinguish the different mutant phenotypes, we performed hierarchical clustering of the average feature values across all single cells labelled in each phenotype's training set (Fig EV1A) and non-linear dimensionality reduction using t-SNE (Maaten & Hinton, 2008) on the training set feature vectors (Fig EV1B). We then used the labelled dataset to train a 2-hidden-layer fully connected neural network (2NN) for each of the endocytic markers. For each single cell, the marker-associated 2NN estimated the probability of each phenotype and we assigned each cell the phenotype with the highest probability. The average classification accuracy on held-out data across all markers and phenotypes was 88.4%, and 18 of the 21 phenotypes had an average classification accuracy > 80% (Fig EV2A and B, Table EV1, see Materials and Methods).

Statistical analyses validated the quality of our pipeline, confirming reproducibility and accuracy of the single-cell phenotypic classifications (see Materials and Methods; Fig EV2C and D). Applying our 2NN to the entire dataset allowed us to accurately detect even a small fraction of aberrant cells, enabling quantification of the variety and penetrance of mutant phenotypes associated with a given mutation (see below).

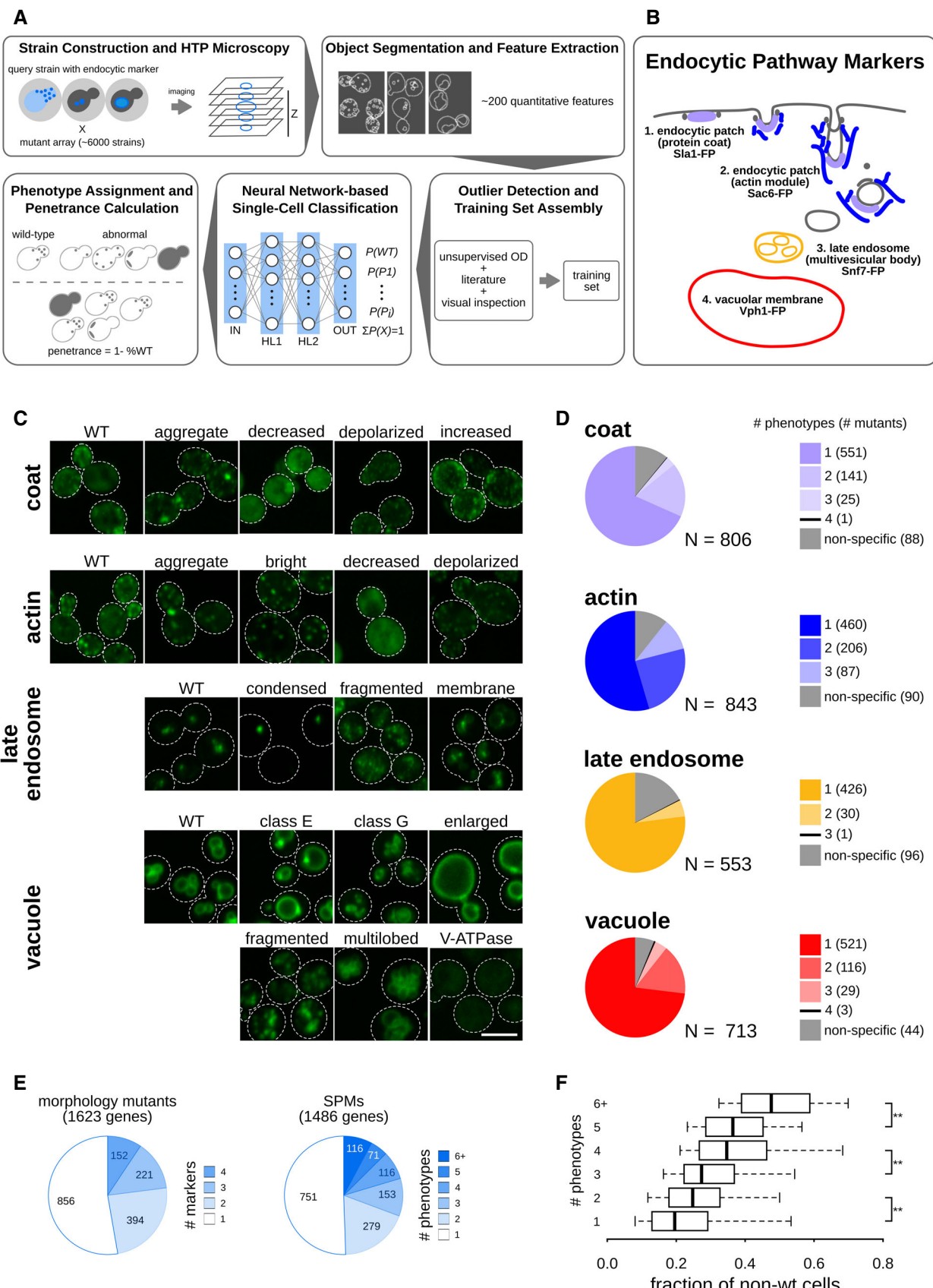

**Figure 1.**

◄ **Figure 1. Twenty-one subcellular endocytic phenotypes identified using computational analysis of single-cell images (see also Figs EV1 and EV2, Tables EV1 and EV2).**

A Diagram of the experimental and computational workflow. Yeast mutant arrays harbouring fluorescently tagged proteins marking specific endocytic compartments were constructed using the synthetic genetic array (SGA) method and imaged using automated high-throughput microscopy. Image and data pre-processing steps included object segmentation and feature extraction, low-quality object clean-up and data standardization. Positive controls and classification training sets were used to train a fully connected 2-hidden-layer neural network (2NN), allowing assignment of phenotypes at the single-cell level and calculation of penetrance.

B Illustration of the endocytosis process and compartment markers. The four endocytic compartment markers used in this study are indicated: Sla1 as a marker of the protein coat component of the endocytic patch (light purple); Sac6 as a marker of the actin component of the endocytic patch (blue); Snf7 as a marker of the late endosome (orange); and Vph1 as a marker of the vacuolar membrane (red). The colours chosen for each marker are used throughout this study. FP: fluorescent protein.

C Example micrographs of yeast cells for each of the 21 subcellular endocytic phenotypes identified in this study. The relevant markers are listed to the left of the micrographs. Dashed lines indicate cell outlines. Scale bar: 5 μm.

D Pie charts showing the proportion of specific phenotype mutants (SPMs) that have one or more distinct aberrant phenotypes, and non-phenotype-specific mutants for each of the compartments screened.

E Pie charts showing the proportion of mutant strains that are morphology mutants for one or more markers (left) and specific phenotype mutants (SPMs) that cause one or more aberrant morphological phenotypes (right). The number of mutants in each category is listed within each section.

F Box plot illustrating the distribution of the fraction of non-wild-type cells for specific phenotype mutants grouped by the number of phenotypes they cause. ** denotes a significant difference between two groups ($P < 0.01$; significance was determined using analysis of variance (ANOVA) with a post hoc Bonferroni test). Central lines represent the median. The number of specific phenotype mutants in each group ranges from 71 to 751. See Table EV2 for details. Whiskers extend to the 5th and 95th percentile.

## Hundreds of yeast genes affect endocytic compartment morphology

To capture the spectrum of phenotypes associated with each mutant strain, we determined the fraction of cells in a mutant strain population that displayed each of the 21 phenotypes using our classifiers described above (Fig 1C). We called a strain a specific phenotype mutant (SPM) if the fraction of cells assigned an aberrant phenotype was significantly greater than that assigned the same phenotype in a control wild-type strain population (see Materials and Methods). In total, we identified 1,486 mutants as SPMs (Fig 1D), with many mutants classified as SPMs for more than one phenotype. We defined a subset of 363 mutants as stringent SPMs, as they had a relatively larger fraction of cells with a specific defect (see Materials and Methods). We also identified a small set of non-phenotype-specific mutants (137 unique genes; Fig 1D) which showed a significant increase in the total percentage of the cell population displaying an aberrant phenotype for a given compartment, even if none of the individual phenotype fractions were high enough for a given strain to be classified as an SPM. In total, we identified 1,623 yeast genes (~30% of screened ORFs) that affect the morphology of one or more endocytic compartments (referred to as morphology mutants; Table EV2; https://thecellvision.org/endocytosis). Thus, yeast endocytosis is remarkably sensitive to single-gene perturbation, consistent with previous siRNA screens in mammalian cells (Collinet *et al*, 2010).

We next examined the extent of morphological pleiotropy, which we define as occurring when a mutant has two or more aberrant morphological phenotypes. It is important to note that morphological pleiotropy does not necessarily imply functional pleiotropy, where a gene affects multiple functionally distinct processes (Paaby & Rockman, 2013). For each marker, some of the morphology mutants showed multiple phenotypes (Fig 1D). Overall, approximately half of the 1,623 morphology mutants showed aberrant phenotypes with more than one of the four markers screened, and approximately half of the SPMs displayed more than one of the 17 aberrant phenotypes (Fig 1E), indicating that morphological pleiotropy is prevalent within this conserved pathway and that numerous genes impinge on multiple stages of endocytosis. The most

pleiotropic mutants (those causing six or more specific phenotypes; 116 SPM genes) were involved in vesicle organization, exocytosis, protein lipidation and membrane fusion. Genes associated with multiple morphological outcomes tended to affect a larger fraction of the cell population (Fig 1F). Morphology mutants were also enriched for TS alleles of essential genes (Fig EV3A and B) and the fraction of essential gene mutants increased with the number of morphological phenotypes (Fig EV3C). However, morphological pleiotropy was not confined to essential genes. For example, mutants of both the essential exocyst complex and the non-essential ESCRT complexes led to phenotypic defects spanning the early and late endocytic compartments (Table EV2).

Genes annotated with roles in a wide range of functions appear to impinge on the endocytic pathway. Only 286 (~18%) of the identified mutant genes were annotated to GO Slim biological process terms associated with endocytosis and the endomembrane system (Table EV2). Similarly, while morphology mutants were enriched for genes conserved between yeast and human (~40% of conserved morphology mutants compared to ~26% on the array, $P < 0.0001$; Fig EV3D), this enrichment was not due to known endocytosis machinery components (Fig EV3E), but included genes involved in a range of bioprocesses, such as DNA replication and repair, transcription and splicing.

## Automated image analysis identifies the spectrum of possible endocytic compartment morphologies

Of the 17 aberrant morphological phenotypes associated with the four endocytic markers, 15 correspond to previously described phenotypes (Table EV1). The unsupervised outlier detection analysis identified two novel phenotypic groups: mislocalization of the late endosomal marker to the vacuolar membrane ("late endosome: membrane" in Fig 1C) and a previously unappreciated vacuolar mutant phenotype characterized by small vacuoles and increased cytosolic localization of the vacuolar marker, Vph1. Because most of the SPMs in this class were genes involved in various aspects of Golgi vesicle transport, we refer to this vacuolar morphology as the "class G" phenotype. We confirmed that the class G was a distinct phenotype, and not an intermediate stage of

one of the known vacuolar phenotypes, by imaging in a 24-h time course at 37°C (Fig EV4A). Since Golgi vesicle transport affects trafficking pathways to the vacuole, the class G phenotype could be a consequence of abnormal vacuolar membrane composition that leads to defects in vacuole formation or membrane fusion and fission.

Comparisons to a panel of gene attributes (Fig EV4B, Table EV3) revealed that morphology mutants in all four compartments were enriched for the same set of features: high conservation across different species, ample genetic interactions (GIs) and protein–protein interactions (PPIs), pleiotropy and multifunctionality, fitness defects and tendency to act as phenotypic capacitors.

Mutants with aberrant phenotypes were often enriched in multiple bioprocesses, both closely related and apparently unrelated to the compartment associated with the aberrant phenotype, suggesting that multiple mechanisms can lead to a particular phenotype (Fig 2, Table EV3). For example, a decrease in actin patch numbers could be due to defects in mRNA processing and transcription, DNA replication and repair, exocytosis or the cell cycle (Table EV3). Stringent SPMs were enriched for more specific protein complexes and biological pathways, which may be suggestive of the mechanisms underlying their aberrant morphological phenotypes (Table EV3). Phenotypes that occur in a relatively high fraction of the population in wild-type strains, such as depolarized patches or multilobed vacuoles (Fig 2), may result from a general cellular response to different stress conditions (environmental or genetic) and tend to be associated with a larger number of SPMs (Fig 2).

The comparison of SPMs for several markers allowed us to search for new connections both within and between the endocytic compartments. We found that different morphology defects can be enriched among genes with roles in the same bioprocesses (Table EV3), possibly reflecting a common biological mechanism. To better understand the relationships between different phenotypes, we measured the pairwise correlations between each of the 17 mutant phenotypes across all SPMs (Fig EV4C, Table EV5). As expected, this comparison revealed a large number of correlated phenotype pairs; however, 86 phenotype pairs (of the 136 possible) were either not significantly correlated or were anti-correlated (Table EV5), suggesting orthogonal or opposite cellular events. For example, enlarged and multilobed vacuoles are anti-correlated, consistent with defects in either membrane fission or fusion. We next evaluated whether pairs of phenotypes shared more stringent SPMs than expected by chance (Table EV5). Of the 136 possible phenotype pairs, 36 pairs shared a significantly ($P < 0.05$, FDR < 0.2) overlapping set of causative gene mutations, and for 15 of these pairs, the overlapping set was enriched in specific protein complexes (Fig 3A, Table EV5). This conservative analysis identified a core set of 13 protein complexes that affect endocytic compartment morphology at multiple levels, including several protein complexes with well-characterized roles in vesicle trafficking (Fig 3B), such as the HOPS, the vacuolar t-SNARE and the retromer complexes, which

are involved in anterograde and retrograde trafficking between the Golgi, endosomes and the vacuole. Mutants of these complexes have defects at the late endosome-vacuole fusion step, or defects in recycling leading to depletion of sorting machinery components, resulting in multiple "fragmented" endosomes (Balderhaar & Ungermann, 2013; Ma & Burd, 2019). Some of the related endocytic morphology defects are likely sequential, while others may stem from independent events. For example, mutations in genes encoding components of the ESCRT complexes caused three connected phenotypes: coat aggregates, condensed late endosomes and class E vacuoles. Defects in ESCRT complex assembly and MVB formation lead to accumulation of cargo at the late endosome—all three phenotypes therefore mark an exaggerated prevacuolar endosome-like compartment (Coonrod & Stevens, 2010). In contrast, mutation of genes encoding general transcriptional regulators such as TFIIH and the core mediator caused pleiotropic endocytic phenotypes which may reflect a series of independent defects in transcription. Another core complex with effects on multiple endocytic compartments is the functionally conserved Dsl1 multisubunit tethering complex, a resident ER complex involved in retrograde Golgi-to-ER trafficking (Andag *et al*, 2001; Reilly *et al*, 2001). As the upstream step in many intracellular vesicle trafficking pathways, disruption of ER-Golgi trafficking can alter both sorting through the secretory/exocytic and Golgi-to-endosome pathways, affecting both early and late endocytic compartments.

To explore the extent to which the 17 aberrant endocytic compartment morphologies translate into a defect in endocytic internalization, we compared our list of SPMs with the published results of a quantitative assay for endocytic recycling of the non-essential gene-deletion collection, based on a GFP-Snc1-Suc2 chimeric protein (Burston *et al*, 2009). All sets of SPMs derived from the 17 aberrant phenotypes were associated with a decrease in endocytic internalization ($P < 0.01$; Fig EV4D, Table EV6), with the exception of SPMs for the vacuolar class G phenotype, which were mostly essential genes (58/79 SPMs, including all of the stringent SPMs, Fig EV3B). We next tested each phenotype class to determine whether the mutants with a more penetrant version of the phenotype were more likely to have an endocytic internalization defect. We compared the defect levels of stringent SPMs to non-stringent SPMs and found a significant increase in defects for four phenotypes: decreased number of actin patches, coat aggregate, condensed late endosome and class E vacuole (Fig EV4D, Table EV6). These phenotypes are likely directly linked to an endocytic internalization defect. The internalization defects of the stringent SPMs for actin patches were the highest of the four compartments (Tables EV2 and EV6). The actin module is the driving force in endocytic internalization and studies have previously shown that mutants with a reduced number of actin structures have defective endocytosis (Weinberg & Drubin, 2012). The remaining three phenotypes linked with internalization defects were those associated with defects in ESCRT complex and MVB formation.

**Figure 2. The spectrum of endocytic compartment morphologies: properties of 17 mutant phenotypes. (see also Fig EV3, Table EV3).**
Representative images of wild-type and mutant cells organized by marker and phenotype (labels on the left of each panel). For each phenotype, three cells labelled for the training set (labelled single cells) and three cells identified by the 2NN classifier (identified single cells) are shown. The table to the right of the images shows (from left to right): (i) the occurrence of each phenotype in a wild-type population (% in WT); (ii) the number of specific phenotype mutants (all) and stringent specific phenotype mutants (str) for each of the 17 mutant morphologies; (iii) the most significantly enriched GO Slim biological process; and (iv) the most significantly enriched protein complex. # denotes term below statistical significance.

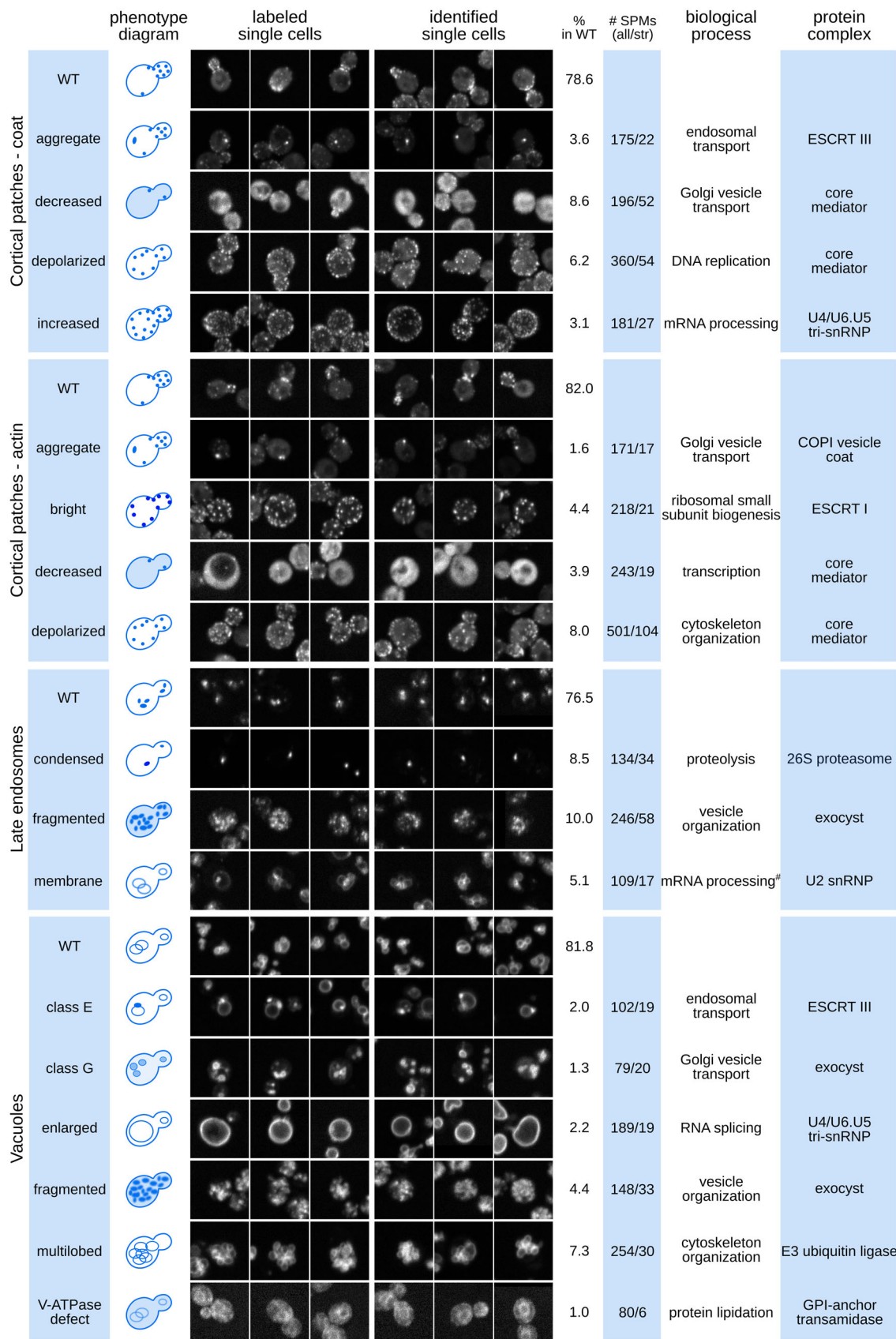

Figure 2.

## Subcellular morphology information and phenotype profiles support prediction of gene function

For virtually all 17 aberrant morphological phenotypes, we found several genes that had not been previously linked to the assessed morphological defects, including ~130 morphology mutants corresponding to largely uncharacterized genes. For example, *YDL176W* caused a decrease in the number of actin patches and concomitant increase in the number of coat patches when mutated. This suggests a defect in actin patch assembly that causes a delay in patch

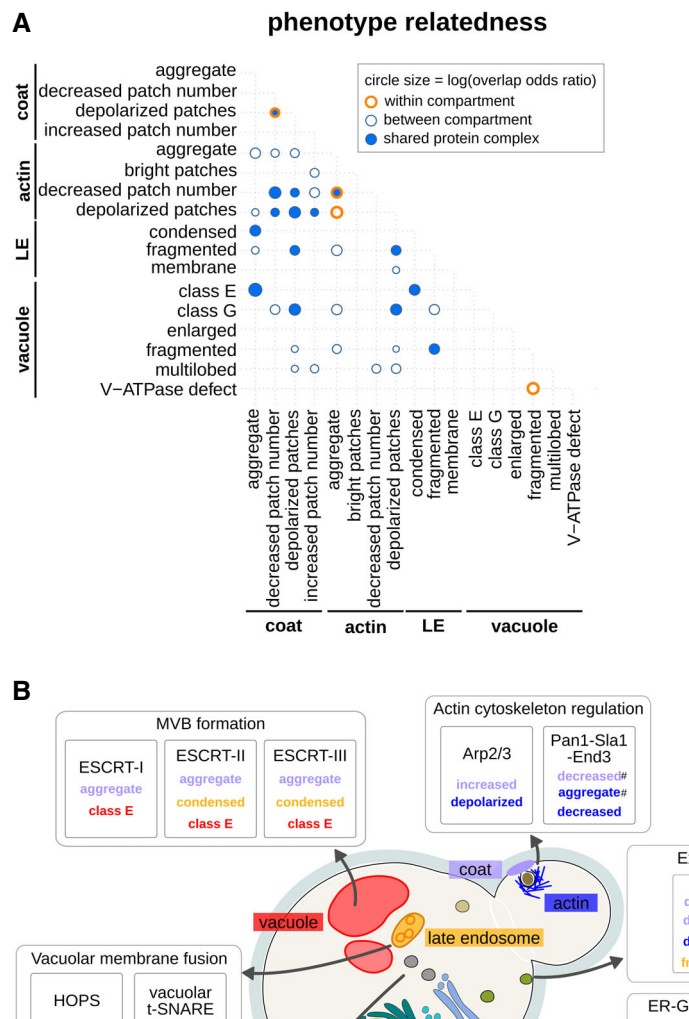

**Figure 3. Analysis of the common morphology mutants of endocytic compartment phenotypes and the relationship to known protein complexes (see also Table EV5).**

A Matrix showing significant overlap of stringent specific phenotype mutants ($P < 0.05$; significance was determined using Fisher's exact tests). Circle size corresponds to the log value of the overlap odds ratio. Orange circles denote same-compartment phenotype pairs. Dark blue fill colour indicates phenotype pairs with at least one enriched protein complex in the overlapping set. LE: late endosome.

B Diagram illustrating co-occurrence of endocytic morphology phenotypes associated with protein complex perturbation. Shown are significant protein complexes from (A) with biological processes and linked phenotype pairs. # denotes a phenotype pair without significant enrichment. Phenotype names are colour-coded by endocytic marker, using the colour key described in Fig 1 and indicated on the yeast cell diagram.

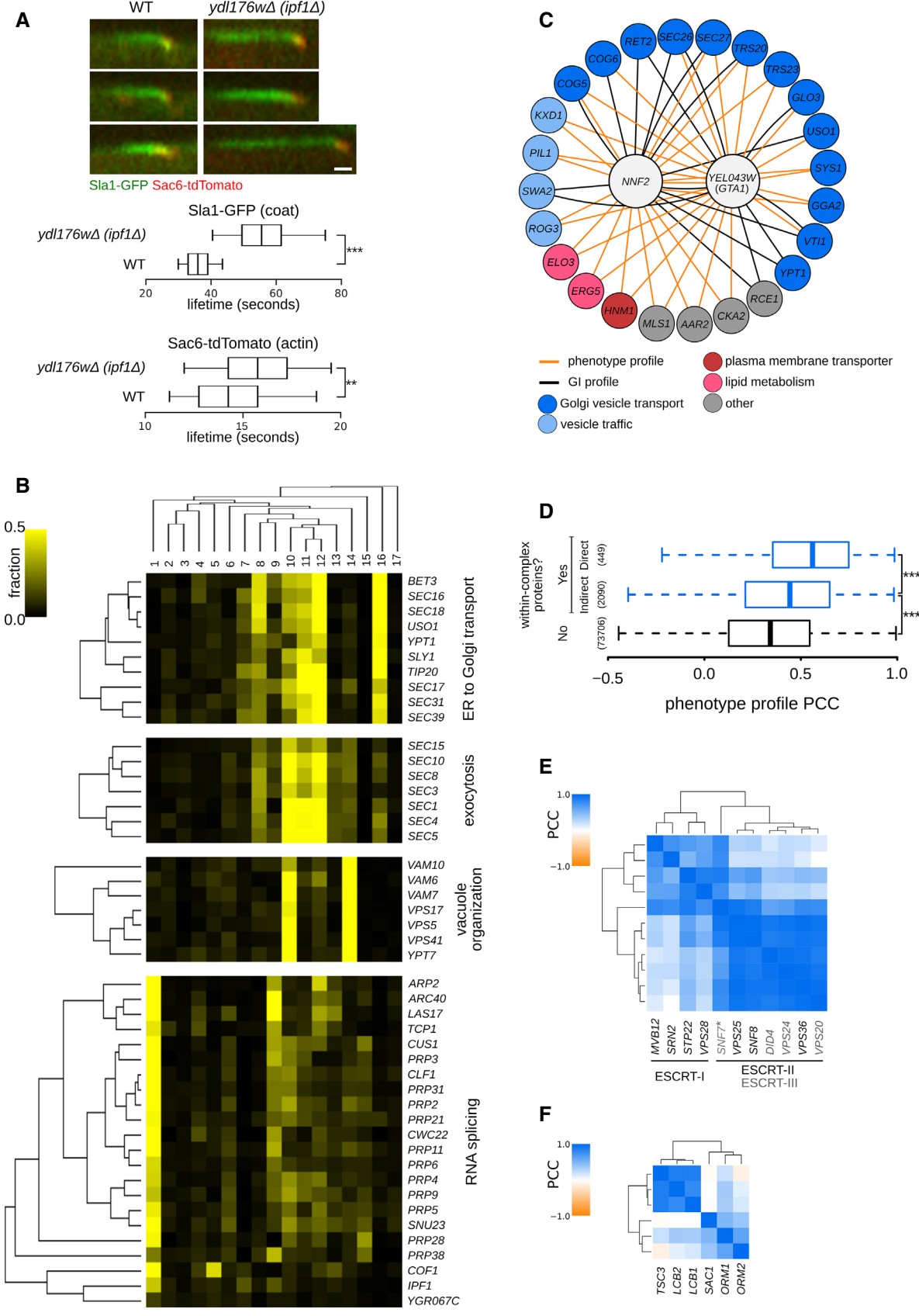

**Figure 4.**

**Figure 4.  Predicting gene function from phenotype profiles (see also Fig EV4).**

A   Endocytic patch formation dynamics in the *ydl176wΔ* (*ipf1Δ*) strain. Patch dynamics were examined using time-lapse fluorescence microscopy of wild-type (WT) and *ipf1Δ* deletion strains carrying reporters for the coat (Sla1–GFP; green) and actin (Sac6-tdTomato; red) modules. Upper: Representative kymographs for the WT and *ipf1Δ* strains. Scale bar: 10 s. Lower: Box plot illustrating the distribution of lifetimes of Sla1-GFP and Sac6-tdTomato patches. The box represents IQR (interquartile range). Whiskers extend to the 5$^{th}$ and 95$^{th}$ percentile. Central lines represent the median. At least 100 patches were analyzed per strain and marker. \*\*, \*\*\*denote a statistically significant difference between the two groups (*P* < 0.01 or *P* < 0.001). Significance was determined using unpaired *t*-tests.

B   Examples of gene clusters obtained with hierarchical clustering of phenotype profiles composed of the 17 specific phenotype fractions. Phenotypes 1–17: [1] coat: increased patch number; [2] coat: aggregate; [3] vacuole: class E; [4] late endosome: condensed; [5] actin: bright patches; [6] late endosome: membrane; [7] actin: aggregate; [8] coat: decreased patch number; [9] actin: decreased patch number; [10] late endosome: fragmented; [11] coat: depolarized patches; [12] actin: depolarized patches; [13] vacuole: multilobed; [14] vacuole: fragmented; [15] vacuole: enlarged; [16] vacuole: class G; [17] vacuole: V-ATPase defect.

C   Interaction network of *NNF2* and *YER043W* (*GTA1*). Genes with phenotype profiles with a correlation > 0.7 and genetic interaction profiles with a correlation > 0.2, and at least two significant correlations to *NNF2* and/or *GTA1* were included in the network.

D   Analysis of phenotype profile similarity between mutants in genes encoding proteins in same or different protein complex structures. Box plot indicates distribution of PCCs between pairs of phenotype profiles for genes that encode protein pairs in direct contact in a protein complex experimental structure (Yes - Direct), code for protein pairs in the same protein complex structure but not in direct contact (Yes - Indirect) and code for protein pairs that do not belong to the same protein complex structure (No). The box represents IQR (interquartile range). Whiskers are Q1-1.5\*IQR and Q3+1.5\*IQR. Central lines represent the median. The number of pairs evaluated in each set is shown on the left side. Significance was determined using one-sided Mann–Whitney *U*-tests. \*\*\**P* < 0.001.

E, F  Differentiation of functionally related protein complexes and protein complex organization using phenotype profiles. Heatmaps showing PCCs between components of the ESCRT complexes (E) and the SPOTS complex (F). A more intense blue colour indicates a higher PCC (scale bar at the top left of each heat map).

internalization and accumulation of upstream components. Indeed, a *ydl176wΔ* mutant harbouring Sla1-GFP and Sac6-tdTomato markers exhibited a 55% increase in the lifetime of Sla1-GFP patches (*P* < 0.0001) and a modest but significant increase in the lifetime of Sac6-tdTomato (7.6% increase, *P* = 0.0012) (Fig 4A). Moreover, the *YDL176W* deletion mutant has an endocytic internalization defect (Burston *et al*, 2009), and *YDL176W* shows a strong negative GI with *SLA2* (Costanzo *et al*, 2016), which encodes an adapter protein that links actin to clathrin and endocytosis. We thus named the *YDL176W* open reading frame *IPF1* for involved in actin patch formation.

As we have shown, half of our SPMs affect multiple compartments and some lead to phenotypes that are present only in a small fraction of the population. To facilitate functional prediction for these genes, we used a multivariate approach that considers all the morphology phenotype classes. For each mutant strain, we assembled a phenotype profile composed of the fraction of cells with aberrant morphology for each of the 17 mutant classes and computed the similarity of phenotype profiles between each pair of morphology mutant genes. Functionally related gene pairs exhibited significantly higher phenotype profile similarities, indicating that phenotype profiles were predictive of a functional relationship (Fig EV5A). Hierarchical clustering of phenotype profiles identified clusters enriched in functionally related genes, including clusters of genes involved in ER to Golgi transport, vacuole organization and exocytosis (Fig 4B). Interestingly, one cluster contained genes encoding regulators of actin and RNA splicing. Unlike most yeast genes, many actin regulatory genes, such as *COF1* and *ARP2*, as well as *ACT1* itself, contain introns and thus depend on mRNA splicing to produce functional proteins and normal regulation of actin cytoskeleton organization (Fig 4B). The same cluster also includes the newly named *IPF1* gene (see above), additionally linking its function to actin cytoskeleton regulation.

Two poorly characterized genes, *YEL043W* and *NNF2* (*YGR089W*), had highly correlated phenotype profiles (PCC = 0.89) that were most similar to profiles of genes involved in Golgi vesicle and endosomal transport (Figs 4C and EV5B). Both gene products are localized to the ER (Huh *et al*, 2003; Chong *et al*, 2015; Kraus *et al*, 2017) and contain coiled-coil domains that are often associated with vesicle tethering proteins (Cheung & Pfeffer, 2016). Moreover, the coiled-coil domains of Yel043w and Nnf2 physically interact with each other (Newman

*et al*, 2000; Wang *et al*, 2012) and the GI profiles of *YEL043W* and *NNF2* are both enriched for interactions with genes involved in vesicle trafficking (Costanzo *et al*, 2016), suggesting a possible role for these two proteins in Golgi vesicle trafficking. We named the *YEL043W* gene *GTA1*, for Golgi vesicle trafficking associated.

In addition, many protein complexes that affect at least one of the screened endocytic markers had a high within-complex phenotype profile correlation (Fig EV5C, Table EV7). Phenotype profiles were more similar between components of the same protein complex structure that are in direct contact when compared to those that are not (Fig 4D). In some cases, these profiles were able to differentiate between closely related complexes and between functional subunits of a complex. For example, ESCRT complex mutants led to the vacuolar class E and related phenotypes. Phenotype profiles were able to differentiate between ESCRT-I and ESCRT-II/III components (and to a lesser extent also between ESCRT-II and ESCRT-III components) (Fig 4E). In another example, phenotypic profiles differentiated two distinct functional subunits of the SPOTS complex, involved in sphingolipid homeostasis (Fig 4F). This modularity is consistent with the known biochemistry: the catalytic activity depends on Lcb1 and Lcb2 and is stimulated by Tsc3, whereas Sac1, Orm1 and Orm2 are believed to play regulatory roles (Fig 4F) (Breslow *et al*, 2010).

## Penetrance is an informative indicator of gene function

Besides specific phenotype information, an important output of our single-cell analysis was quantification of penetrance, defined as the total percentage of the population with an aberrant phenotype, in each mutant for each compartment. Among the morphology mutants were 1216 penetrance mutants that had a significant increase in penetrance compared to the control strain (Table EV2). For ~90% of these mutants, the morphology defect was incompletely penetrant (Fig 5A). We binned mutants based on low, intermediate or high penetrance and found that each group of genes was enriched for distinct functions (Table EV8). We previously showed that a network based on genetic interaction profiles provides a global view of the functional organization of the cell (Costanzo *et al*, 2016). Thus, we next examined where these genes localized relative to biological process-enriched clusters on the global genetic

interaction profile similarity network using spatial analysis of functional enrichment (SAFE) (Fig 5B) (Baryshnikova, 2016). Highly penetrant mutants localized to bioprocesses that are closely related to the function of the screened marker, genes corresponding to intermediate penetrance mutants mapped to "neighbouring" processes and low penetrance mutants localized to clusters enriched for more functionally "distant" processes. For example, genes with highly penetrant Snf7-GFP phenotypes reflecting defects in late endosome morphology, mapped to clusters on the global genetic network representing multivesicular body sorting and vesicle trafficking, while genes exhibiting intermediate penetrance were located within vesicle trafficking-, glycosylation- and polarity-enriched network clusters. Finally, low penetrance mutants tend to localize to regions of the global genetic network corresponding to vesicle trafficking, polarity, mRNA processing and transcription (Fig 5B). Thus, penetrance alone is informative about the functional relationship between processes.

## Replicative age, asymmetric inheritance and stress all contribute to incomplete penetrance in an isogenic cell population

Several factors have been suggested to affect penetrance in isogenic populations, including cell cycle position, cell size, replicative age, asymmetric segregation of molecular components, daughter-specific expression and environmental factors (Colman-Lerner et al, 2001, 2005; Avery, 2006; Newman et al, 2006; Henderson & Gottschling, 2008; Levy et al, 2012; Knorre et al, 2018). Our quantitative single-cell analysis of the morphological defects associated with each marker provided a unique opportunity to explore some of the potential molecular and cellular mechanisms underlying incomplete penetrance.

### Replicative age and penetrance

In yeast, replicative age can be assessed by staining chitin-rich bud scars to distinguish mother cells of different ages (Guthrie & Fink, 2002). The average replicative lifespan for our wild-type BY4741 strain is 20–30 generations (Liu et al, 2015; McCormick et al, 2015); thus, old mothers are rare in a cell population. We examined wild-type control cells and five mutants with vacuole defects that had incomplete penetrance, including three mutants (rrd2Δ, cka2Δ and rpl20bΔ) that are known to display a modestly extended replicative lifespan (McCormick et al, 2015), and two vacuole inheritance mutants (vac8Δ and vac17Δ) (Tang et al, 2003). We quantified the amount of bud scar staining in each cell, binned cells roughly corresponding to number of bud scars, thus in bins of unequal size, and assessed whether each cell had a vacuole defect.

For wild-type and all five mutants, the fraction of outliers was lowest in the cells with lowest bud scar staining, corresponding to new daughters, and increased in mother cells with each cell division (Fig 6A, upper panel). In the bin with highest bud scar staining, corresponding to 5+ generations and consisting of ~3% of the population, approximately half of the wild-type cells (53%) and from 51 to 94% of the mutant cells had a vacuolar morphology defect (Fig 6A and B). Thus, aberrant vacuolar morphology increases with the number of cell divisions even in young cells. When compared to wild type, for four mutants (cka2Δ, rpl20bΔ, vac8Δ and vac17Δ), we see that the relative contribution of the gene mutation decreases with each cell division (Fig EV6). This suggests that as cells get

older, age-specific effects may contribute more to penetrance than gene-specific effects.

Much of the work on replicative ageing has been done on old mother cells but more recent studies have identified a number of factors that accumulate in relatively young mothers including oxidized proteins, protein aggregates and reactive oxygen species (Knorre et al, 2018). Multiple studies have reported that cell size increases in old mother cells (Zadrag-Tecza et al, 2009; Janssens & Veenhoff, 2016). We quantified the size of our bud scar-stained cells and confirmed that mother cells increased in size with replicative age, even in their first five generations (Fig 6A, lower panel), with no significant difference in cell size between wild-type cells and the mutants we assayed. In summary, in these experiments, increased penetrance seems to correlate with increased replicative age.

### Asymmetric organelle inheritance and penetrance

Organelle inheritance is an intrinsic component of cell division and mutations that affect this process can lead to cellular heterogeneity. In yeast, VAC8 and VAC17 are required for vacuole movement and partitioning between the mother and daughter cell (Tang et al, 2003). We imaged cells of wild-type and vac17Δ strains, with markers for vacuole and nucleus, stained for bud scars and compared vacuole morphology defects in old and young cells of the two strains (Fig 6C). In these inheritance mutants, multilobed vacuoles were associated with ageing and appeared at a much younger age compared to the wild-type strain, leading to an increase in the fraction of the population that had a vacuolar morphology defect (Fig 6A and C). Thus, the observed cell-to-cell variability in deletion mutants of these two genes is a result of at least two factors: (i) defects in vacuole inheritance where daughter cells do not inherit a vacuole from their mother, but rather have to make one de novo (mother–daughter heterogeneity) and (ii) replicative ageing contributing to the accumulation of vacuole fission products with each cell division cycle, leading to multilobed vacuoles of increasing severity (replicative age-dependent heterogeneity). Similar to these vacuole mutants, asymmetric inheritance of many cellular components could affect penetrance.

### Stress response and penetrance

Exposure to stress can lead to heterogeneous survival rates of isogenic yeast cells (Levy et al, 2012), and can reduce penetrance in Caenorhabditis elegans (Casanueva et al, 2012). Single-cell analysis allowed us to address whether there was any relationship between levels of stress response and penetrance of morphology defects. We examined the unfolded protein response (UPR), which monitors folding of membrane and secreted proteins in the endoplasmic reticulum (Wu et al, 2014). We first compared penetrance mutants with a study that had assayed UPR in the gene-deletion collection using flow cytometry (Jonikas et al, 2009). For actin and coat, but not vacuole, an increased UPR was associated with mutants that had high penetrance in our screens (Table EV8). To explore the relationship between penetrance and the stress response in single cells, we crossed a reporter gene under the control of unfolded protein response elements (UPREs) (Jonikas et al, 2009) into mutants that had incomplete penetrance for actin or vacuole defects (Table EV8). We then measured reporter activity as a proxy for the stress response level in each cell, divided by the cell area to normalize for cell size and quantified penetrance as a function of stress response.

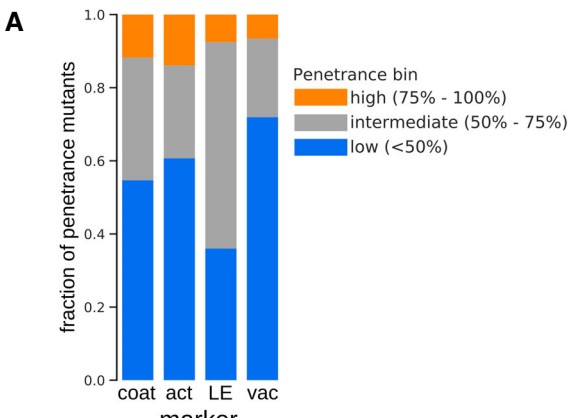

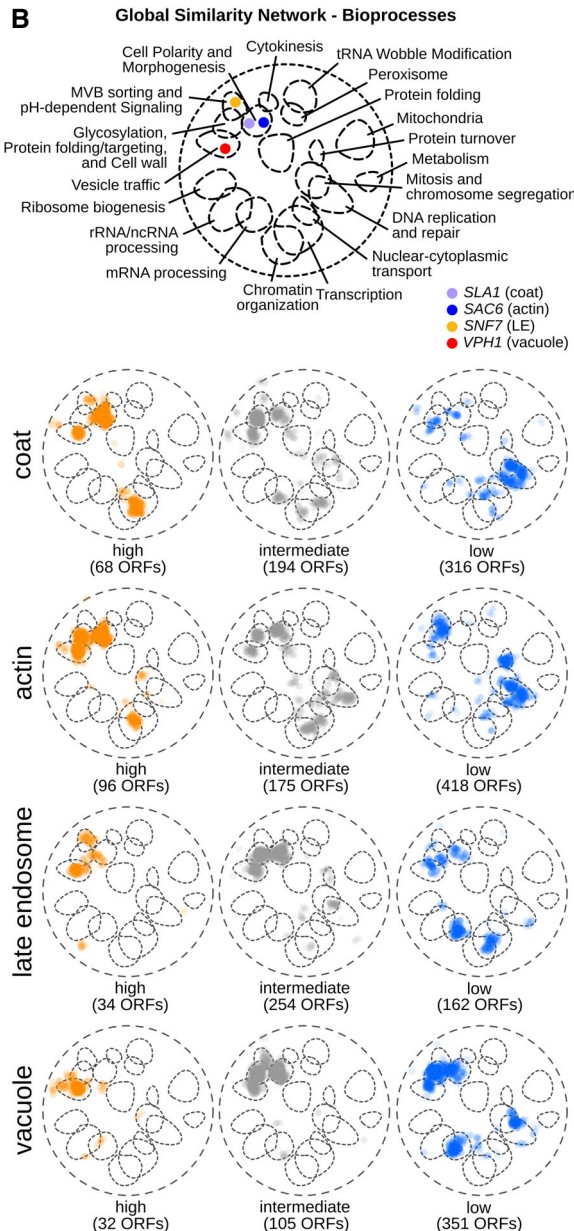

**Figure 5. Functional analysis of incomplete penetrance (see also Table EV8).**

A  Stacked bar graph with fractions of penetrance mutants belonging to each penetrance bin for the four endocytic markers. act: actin; LE: late endosome; vac: vacuole.

B  SAFE (Spatial Analysis of Functional Enrichment) of penetrance mutants grouped according to penetrance. Top: Bioprocess key for interpreting the global similarity network of yeast genetic interactions visualized using SAFE, which identifies regions of the network enriched for specific biological processes (Costanzo *et al*, 2016). Coloured dots denote the localization of the 4 marker genes within the global similarity network. Below: SAFE of penetrance mutants grouped according to their penetrance and marker. Orange: genes whose mutation caused high penetrance; grey: intermediate penetrance genes; blue: low penetrance genes. Numbers in brackets refer to the number of unique ORFs in each group.

The relationships between penetrance and the UPR were different for the two assayed compartments, and the results were consistent with our correlation analysis (Table EV8). For approximately half of the mutants affecting actin, an increased UPR was associated with increased penetrance (Fig 6D, left panel, clusters 1 and 2), while the penetrance of vacuolar morphology was fairly constant across different levels of UPR for most mutants (Fig 6D, right panel, cluster 1). These findings indicate that UPR activation is correlated with penetrance of actin-based endocytosis phenotypes. At the molecular level, the UPR has been proposed to indirectly affect actin cytoskeleton remodelling by activating the cell wall integrity pathway (Bonilla *et al*, 2002; Bonilla & Cunningham, 2003; Levin, 2005), which suggests that the connection between the UPR and actin-based endocytosis phenotypes may be causal.

These experiments show that replicative age, organelle inheritance and response to stress are among the possible factors that contribute to incomplete penetrance in isogenic populations.

## Discussion

To explore how single-cell analysis can be used to assess cell-to-cell variability, morphological pleiotropy and incomplete penetrance, we developed a high-content screening pipeline that allowed us to interrogate sets of yeast mutants for effects on subcellular compartment morphology of a conserved pathway. Using a single-cell-level neural network classifier, we assigned over 16 million cells to one of 21 distinct endocytic phenotypes and obtained penetrance information for four markers for ~5,600 different yeast mutants (corresponding to ~5,300 genes, or ~90% of the genes in the yeast genome). We found that ~1,600 unique yeast genes affect the morphology of one or more endocytic compartments. This dataset provides rich quantitative phenotypic information revealing roles of specific genes in shaping compartment morphology and the functional connections between genes and the compartments they perturb.

We used machine learning to perform outlier detection followed by classification of phenotypes to describe endocytic compartment morphology. These data allowed us to define possible morphologies for several functionally important cell compartments and also to build phenotype profiles, which links all assayed phenotypes associated with a specific genetic perturbation. The resulting phenotypic profiles enabled us to predict gene function and revealed functional information at the level of bioprocesses and protein complexes that was not evident by considering individual phenotypes.

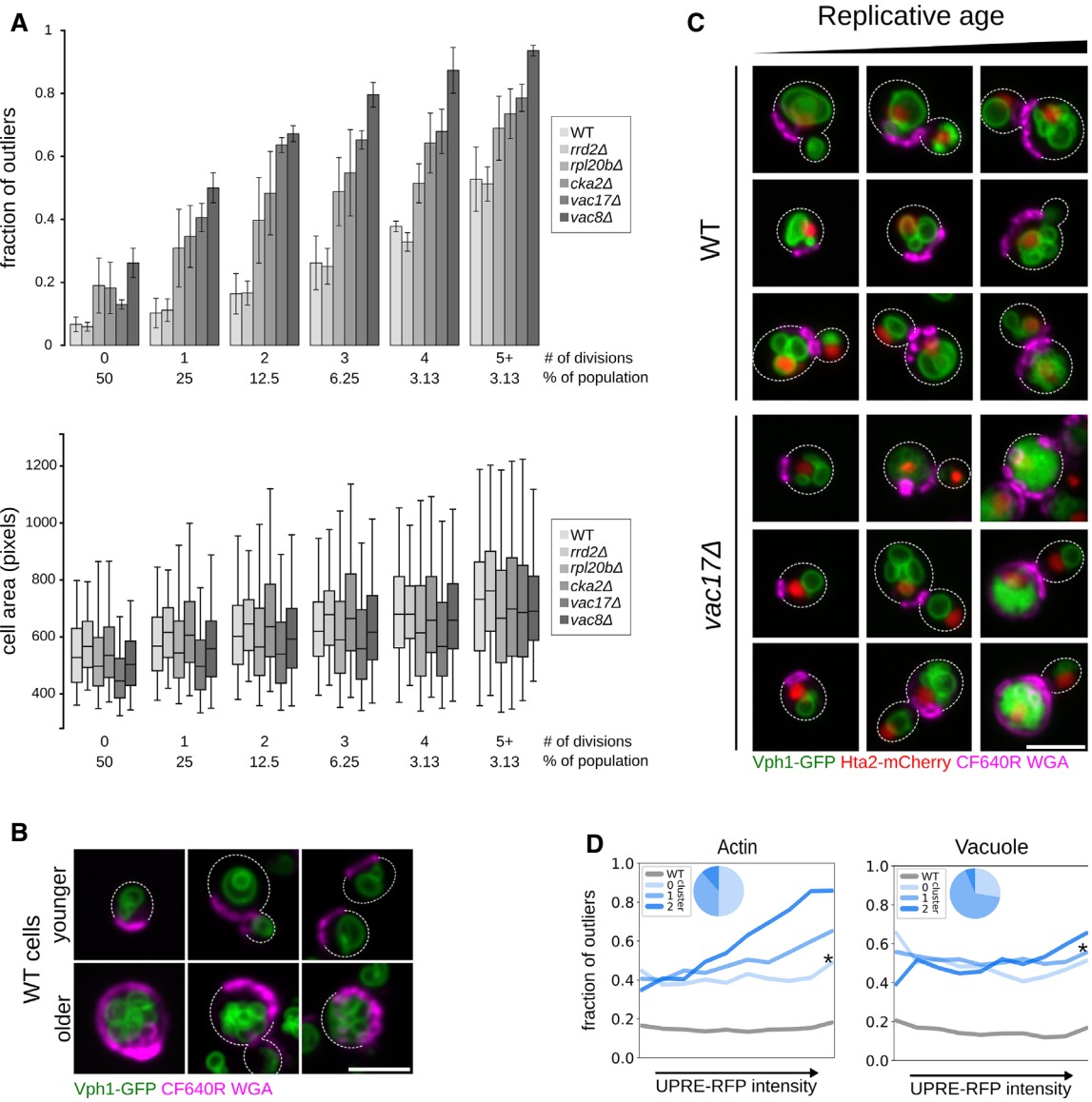

**Figure 6. Factors contributing to incomplete penetrance (see also Table EV8).**

A Penetrance as a function of replicative age. Top: Bar graph showing the fraction of outliers in populations of increasing replicative age (# of divisions) for wild-type (WT), and 5 mutant strains (rrd2Δ, rpl20bΔ, cka2Δ, vac8Δ and vac17Δ). Data are presented as mean of three biological replicates ± SD. Bottom: Box plot with the distribution of cell sizes for the same populations of cells. Central lines represent the median. Whiskers extend to the 5th and 95th percentile. At least 6,800 cells were analyzed per strain (up to 19,500 cells).

B Micrographs of young (top row of images) and older (bottom row of images) wild-type (WT) cells expressing Vph1-EGFP (green vacuole) and stained with CF640R WGA (magenta bud scars). Dashed lines denote cell outlines. Scale bar: 5 μm.

C Combined effect of replicative age and a vacuole inheritance defect on penetrance. Micrographs of wild-type and vac17Δ cells expressing Vph1-EGFP (green vacuole) and Hta2-mCherry (red nucleus), stained with CF640R WGA (magenta bud scars). Cells with increasing bud scar staining (replicative age) are shown from left to right. Dashed lines denote cell outlines. Scale bar: 5 μm.

D Relationship between stress response and penetrance. Single-cell UPRE-RFP levels were measured in ~60 different mutant strains that we had identified as penetrance mutants with intermediate penetrance with defects in actin or vacuole morphology. Cells were binned into equal-sized bins, from low to high stress response, assessed as outlier or inlier and clustered based on their penetrance profile (composed from the fraction of outliers in each stress response bin). Each line plot represents a penetrance profile. * denotes the cluster with a profile most similar to wild type. Inset pie charts show the proportion of mutant strains in each cluster.

Our analysis focused on markers that report on endocytosis, but the combined experimental and computational pipeline that we describe can be readily extended to unrelated markers and phenotypes, enabling broader functional resolution. At this stage, the budding yeast system remains ideally suited to a large morphological survey of subcellular compartment morphology, given the availability of arrayed reagents for assessing loss- and gain-of-function perturbations in both essential and non-essential genes, and the

ease of live-cell imaging of strains carrying fluorescent markers (Ohya *et al*, 2015; Mattiazzi Usaj *et al*, 2016). No matter the system used, a systematic analysis of phenotype profiles will greatly enhance our understanding of cellular function and lead to a more refined hierarchical model of the cell.

The rich phenotype information associated with single-cell images enables the precise quantification of the prevalence of morphological phenotypes in a given cell population. We discovered that both incomplete penetrance, in which only a fraction of cells in a population have a mutant phenotype, and morphological pleiotropy, in which a specific mutation causes several phenotypically distinct subpopulations, are prevalent among mutant strains with defects in endocytic compartment morphology. More than half of the morphology mutants we identified showed aberrant phenotypes for more than one of the four screened compartments, with the most pleiotropic mutants (those causing six or more specific phenotypes) being the most penetrant. Systematic analysis allows us to begin to explore the biological relevance and mechanisms of variable penetrance. For example, we were able to associate specific bioprocesses with high and low penetrance mutants and to identify a number of protein complexes whose mutation is associated with morphological pleiotropy.

Studies in yeast and mammalian cell systems have begun to address cellular heterogeneity using single-cell transcriptomics to identify subpopulations of cells in specific states, such as cancer, or during the cell cycle, cell differentiation and exposure to stress (Patel *et al*, 2014; Buettner *et al*, 2015; Dixit *et al*, 2016; Marques *et al*, 2016; Gasch *et al*, 2017). Others have used cell imaging techniques to quantify both the structural and spatio-temporal properties of complex biological systems at the single-cell level (Bakal *et al*, 2007; Loo *et al*, 2007; Liberali *et al*, 2014; de Groot *et al*, 2018). Regardless of the read-out, phenotypic heterogeneity appears to be a general feature of cell populations, and so far, most studies have not directly addressed the biology underlying incomplete penetrance. Our ability to systematically assess single-cell phenotypes in mutant cell arrays enabled us to show that replicative age, asymmetric organelle inheritance and stress response all contribute to the incomplete penetrance of single-gene mutations.

A number of other deterministic and regulated factors, such as noise in biological systems, micro-environment, epigenetic regulation, and the lipid and metabolic state of the cell have the potential to affect the penetrance and expressivity of a trait. In fact, for the majority of mutants, variability in morphological phenotypes between individual cells in an isogenic cell population is likely not driven solely by a genotype-to-phenotype relationship, but rather by a combination of smaller contributions from various effects that impact single cells differently depending on their physiological state. A deeper understanding of this variability may also have broad medical implications and should provide insight into the variable penetrance of genes affecting developmental programs and disease genes (Cooper *et al*, 2013; Kammenga, 2017; Li *et al*, 2019; Xavier da Silveira Dos Santos & Liberali, 2019).

# Materials and Methods

### Reagents and Tools table

| Reagent/Resource | Reference or source | Identifier or catalog number |
|---|---|---|
| **Experimental models** | | |
| *Saccharomyces cerevisiae*: DMA#, DMA-SLOW# <br> *MAT**a** xxxΔ::KANMX his3Δ1 leu2Δ0 ura3Δ0 met15Δ0* | Yeast Deletion Collection (Giaever *et al*, 2002) | N/A |
| *S. cerevisiae*: TSA# <br> MATa xxx-ts::KANMX *his3Δ1 leu2Δ0 ura3Δ0 met15Δ0* | Yeast Collection of Temperature-sensitive Strains (Li *et al*, 2011; Costanzo *et al*, 2016) | v6.0 |
| *S. cerevisiae*: BY4741 <br> MATa *his3Δ1 leu2Δ0 ura3Δ0 met15Δ0* | Brachmann *et al* (1998) | ATCC: 9483801 |
| *S. cerevisiae*: Y7092 <br> *MATα can1Δ::STE2pr-Sp_his5 lyp1Δ his3Δ1 leu2Δ0 ura3Δ0 met15Δ0* | (Tong & Boone, 2006) | N/A |
| *S. cerevisiae*: Y8835 <br> *MATα ura3Δ::NATMX; can1Δ::STE2pr-Sp_his5 lyp1Δ his3Δ1 leu2Δ0 ura3Δ0 met15Δ0* | Costanzo *et al* (2010) | N/A |
| *S. cerevisiae*: BY5841 <br> *MATα VPH1-GFP::HIS3 HTA2-mCherry::NATMX can1pr::RPL39pr-tdTomato::CaURA3::can1Δ::STE2pr-LEU2 lyp1Δ his3Δ1 leu2Δ0 ura3Δ0* | This study | N/A |
| *S. cerevisiae*: BY6285 <br> *MATα SAC6-yEGFP::NATMX ura3Δ0::URA3::UPRE-CYC1pr-mCherry can1Δ::STE2pr-Sp_His5 lyp1Δ leu2Δ0 his3Δ1 ura3Δ0 met15Δ0* | This study | N/A |

**Reagents and Tools table** (continued)

| Reagent/Resource | Reference or source | Identifier or catalog number |
|---|---|---|
| *S. cerevisiae*: BY6279<br>*MATα VPH1-yEGFP::NATMX ura3Δ0::URA3::UPRE-CYC1pr-mCherry can1Δ::STE2pr-Sp_His5 lyp1Δ leu2Δ0 his3Δ1 ura3Δ0 met15Δ0* | This study | N/A |
| *S. cerevisiae*: Y15247<br>*MATα VPH1-yEGFP::NATMX SLA1-tdTomato::URA3 can1Δ::STE2pr-Sp_his5 lyp1Δ his3Δ1 leu2Δ0 ura3Δ0 met15Δ0* | This study | N/A |
| *S. cerevisiae*: Y15248<br>*MATα SLA1-yEGFP::NATMX SAC6-tdTomato::URA3 can1Δ::STE2pr-Sp_his5 lyp1Δ his3Δ1 leu2Δ0 ura3Δ0 met15Δ0* | This study | N/A |
| *S. cerevisiae*: Y15249<br>*MATα SLA1-yEGFP::NATMX SNF7-tdTomato::URA3 can1Δ::STE2pr-Sp_his5 lyp1Δ his3Δ1 leu2Δ0 ura3Δ0 met15Δ0* | This study | N/A |
| *S. cerevisiae*: Y15250<br>*MATα SNF7-yEGFP::NATMX VPH1-tdTomato::URA3 can1Δ::STE2pr-Sp_his5 lyp1Δ his3Δ1 leu2Δ0 ura3Δ0 met15Δ0* | This study | N/A |
| *S. cerevisiae*: Y15251<br>*MATα VPH1-yEGFP::NATMX can1Δ::STE2pr-Sp_his5 lyp1Δ his3Δ1 leu2Δ0 ura3Δ0 met15Δ0* | This study | N/A |
| *S. cerevisiae*: Y15252<br>*MATα SNF7-yEGFP::NATMX can1Δ::STE2pr-Sp_his5 lyp1Δ his3Δ1 leu2Δ0 ura3Δ0 met15Δ0* | This study | N/A |
| *S. cerevisiae*: Y15253<br>*MATα SAC6-yEGFP::URA3 can1Δ::STE2pr-Sp_his5 lyp1Δ his3Δ1 leu2Δ0 ura3Δ0 met15Δ0* | This study | N/A |
| *S. cerevisiae*: Y15254<br>*MATα SLA1-yEGFP::NATMX can1Δ::STE2pr-Sp_his5 lyp1Δ his3Δ1 leu2Δ0 ura3Δ0 met15Δ0* | This study | N/A |
| *S. cerevisiae*: Y15255<br>*MATα VPH1-tdTomato::URA3 can1Δ::STE2pr-Sp_his5 lyp1Δ his3Δ1 leu2Δ0 ura3Δ0 met15Δ0* | This study | N/A |
| *S. cerevisiae*: Y15256<br>*MATα SAC6-tdTomato::URA3 can1Δ::STE2pr-Sp_his5 lyp1Δ his3Δ1 leu2Δ0 ura3Δ0 met15Δ0* | This study | N/A |
| **Oligonucleotides** | | |
| Primer: URA3pr-F:<br>CAAAGAAGGTTAATGTGGCTGTGGTTTCAGGGTCCATAAAGCTTTTCAATTCATCATTTTTTTTTTATTCTTTTTTTTGATTTCGG | This study | N/A |
| Primer: dn_mCherry-R:<br>CTGTTACTTGGTTCTGGCGAGGTATTGGATAGTTCCTTTTTATAAAGGCCCCTCGAGGTCGACGGTATCG | This study | N/A |
| Primer: MMU-Sla1-F: CAAGCCAACATATTCAATGCTACTGCATCAAATCCGTTTGGATTCGGTGACGGTGCTGGTTTA | This study | N/A |
| Primer: MMU-Sla1-R: TTGCCATTTTCACGAGTATAAGCACAGATTGTACGAAACTATTTCGATATCATCGATGAATTCG | This study | N/A |
| Primer: MMU-Sac6-F: CGTGCAAGATTAATTATTACTTTTATCGCTTCGTTAATGACTTTGAACAAAGGTGACGGTGCTGGTTTA | This study | N/A |
| Primer: MMU-Sac6-R: CGTATAACGGAGCATTGGAACAAGAAAGCTGAGTAGAAAACAGGTGATATCATCGATGAATTCG | This study | N/A |
| Primer: MMU-Snf7-F: GAAGATGAAAAAGCATTAAGAGAACTACAAGCAGAAATGGGGCTTGGTGACGGTGCTGGTTTA | This study | N/A |
| Primer: MMU-Snf7-R: AGAACACCTTTTTTTTTTTCTTTCATCTAAACCGCATAGAACACGTGATATCATCGATGAATTCG | This study | N/A |
| Primer: MMU-Vph1-F: GACATGGAAGTCGCTGTTGCTAGTGCAAGCTCTTCCGCTTCAAGCGGTGACGGTGCTGGTTTA | This study | N/A |
| Primer: MMU-Vph1-R: GTGGATTGGATTGCAAGTCTAACGTTTTCATGAGATAAGTTTGGCGATATCATCGATGAATTCG | This study | N/A |
| **Recombinant DNA** | | |
| pPM47 | Merksamer *et al* (2008) | Addgene_20132 |
| pKT209 | Sheff and Thorn (2004) | Addgene_8730 |
| pFA6a-link-yEGFP-*NATMX4* | The Boone lab | N/A |
| **Chemicals, enzymes and other reagents** | | |
| Dextran Alexa Fluor 647 | Molecular Probes, Invitrogen | D22914 |
| FM 4-64 | Molecular Probes, Invitrogen | T13320 |
| CF640R WGA | Biotium | 29026 |
| Concanavalin A | MP Biomedicals | 195283 |
| L-Canavanine sulphate salt | Sigma-Aldrich | C9758 |

**Reagents and Tools table**   (continued)

| Reagent/Resource | Reference or source | Identifier or catalog number |
| --- | --- | --- |
| Nourseothricin | Werner BioAgents | CAS 96736-11-7 |
| S-aminoethyl-L-cysteine | Sigma-Aldrich | A2636 |
| Geneticin | Life Technologies | 11811098 |
| High-Speed Plasmid Mini Kit | FroggaBio | PD300 |
| QIAPrep Spin Miniprep Kit | Qiagen | 27106 |
| QIAQuick PCR Purification Kit | Qiagen | 28106 |
| MasterPure Yeast DNA Purification Kit | Epicentre | MPY80200 |
| PerfectTaq Plus MasterMix | 5 Prime | 2200095 |
| Expand High Fidelity PCR System | Roche, Sigma-Aldrich | 11732650001 |
| **Software and Algorithms** | | |
| CellProfiler | https://cellprofiler.org (Carpenter *et al*, 2006) | v2.0 |
| SGATools | http://sgatools.ccbr.utoronto.ca/ (Wagih *et al*, 2013) | N/A |
| STEM: Short Time-series Expression Miner | http://www.cs.cmu.edu/~jernst/stem/ (Ernst & Bar-Joseph, 2006) | N/A |
| TheCellMap | http://thecellmap.org/ (Usaj *et al*, 2017) | N/A |
| ImageJ | https://imagej.nih.gov/ij/ (Schneider *et al*, 2012) | v1.46 or newer |
| Volocity | PerkinElmer | v6.3 |
| **Other** | | |
| BM3 Benchtop System | S&P Robotics | |
| PerkinElmer Opera HCS System | PerkinElmer | |
| PerkinElmer Opera Phenix HCS System | PerkinElmer | |
| DMI 6000B fluorescence microscope with ImageEM CCD camera | Leica Microsystems and Hamamatsu | |

## Methods and Protocols

### Query strain construction and construction of mutant arrays for imaging

To visualize endocytic compartments in living yeast cells, we C-terminally tagged 4 yeast proteins selected to visualize the endocytic compartments of interest with the yeast enhanced green fluorescent protein (yEGFP) or tdTomato. We used the polymerase chain reaction (PCR) to amplify an integration fragment containing: (i) homologous regions 45 bp up- and downstream of the target ORF's C terminus; (ii) the fluorescent protein (FP) ORF; and (iii) the selection marker. Plasmids pKT209 (pFA6a-link-yEGFP-*CaURA3*) (Sheff & Thorn, 2004) and pFA6a-link-tdTomato-*CaURA3* were used as templates. Plasmid pFA6a-link-tdTomato-*CaURA3* was constructed by replacing the yEGFP-*ADH1term* fragment between sites SalI/BglII in pKT209 with the tdTomato-*ADH1term* fragment. Switcher plasmid p4339 was used to exchange the *CaURA3MX4* cassette with the

*NATMX4* resistance cassette to generate yEGFP-*NATMX4*-tagged strains (Tong & Boone, 2007). Primers (starting with MMU_*) used to PCR FP-tagging cassettes for genomic integration are listed in the Reagents and Tools Table. The lithium acetate transformation method was used to introduce the PCR product into yeast cells (Gietz, 2014). The yeast proteins used as markers were as follows: Sac6 for the actin module of actin cortical patches; Sla1 for the coat module of actin cortical patches; Snf7 for late endosomes; and Vph1 for vacuoles. All four proteins have been used previously as markers for these compartments (Kaksonen *et al*, 2005; Teis *et al*, 2008; Zhao *et al*, 2013). *Saccharomyces cerevisiae* strains and oligonucleotides used in the study are listed in the Reagents and Tools Table.

To test for possible growth or other functional defects associated with the fluorescent protein tags, we performed the following tests: (i) staining with FM 4-64 to check for a potential defect in endocytic internalization; (ii) real-time fluorescence microscopy imaging to

check for potential fluorescent tag-effects on Sla1 and Sac6 endocytic patch formation dynamics; (iii) assessment of growth using serial spot dilutions on standard rich YPD media (1% (w/v) yeast extract, 2% (w/v) peptone, 2% (w/v) dextrose, 2% (w/v) agar) at different temperatures; d) mating of constructed FP-tagged query strains with strains carrying mutations in genes that had genetic interactions with *SAC6*, *SLA1*, *SNF7*, or *VPH1*, followed by diploid selection, sporulation, and tetrad dissection to assess the growth of the double mutant progeny. A list of genetic interactions was obtained from Costanzo *et al* (2010, 2016). All of these experiments revealed no effect of the fluorescent tag on the tagged protein's function, except for Snf7-GFP (and Snf7-tdTomato), where we confirmed an effect of the C-terminal fluorescent tag on Snf7p's function, as has been observed previously with all ESCRT-III complex components (Teis *et al*, 2008).

The constructed FP-tagged query strains were crossed to the haploid *MAT***a** deletion collection (Giaever *et al*, 2002) and to a collection of mutant strains carrying temperature-sensitive (TS) alleles of essential genes (Li *et al*, 2011; Costanzo *et al*, 2016). Haploid strains carrying both the fluorescent protein marker and the gene mutation from the mutant strain collections were selected using the SGA method (Tong & Boone, 2006). All SGA selection steps involving a TS allele were conducted at permissive temperature (26°C). All SGA selection steps involving non-essential gene-deletion mutants were conducted at 30°C. Sporulation was conducted at 22°C. For secondary, medium-scale screens, used also to determine penetrance reproducibility, false-positive (FPR) and false-negative rates (FNR), 1,910 strains (36% of the complete array) were chosen from strains with both significant and non-significant phenotype fractions and SGA was done in biological duplicate. Strains included in the secondary array are marked in Table EV2.

### Preparation and imaging of live yeast cells

#### High-throughput microscopy

Yeast cell cultures were prepared for microscopy and imaged as previously described (Chong *et al*, 2015; Cox *et al*, 2016), with some modifications. Briefly, haploid mutant *MAT***a** strains expressing tagged FPs derived from SGA were grown and imaged in low fluorescence synthetic minimal medium (Sheff & Thorn, 2004) supplemented with antibiotics and 2% glucose. Non-essential gene-deletion mutants were grown and imaged in logarithmic phase at 30°C, and TS mutants of essential genes were first grown to mid-logarithmic phase and imaged at 26°C and then incubated for 3 h at 37°C and imaged at 37°C. Cells were transferred to a Concanavalin A (ConA) coated 384-well PerkinElmer CellCarrier Ultra imaging plate and centrifuged for 45 s at 500 rpm before imaging. To aid in cell segmentation, Dextran Alexa Fluor 647 (Molecular Probes) was added to cells in low fluorescence medium to a final concentration of 10 μg/ml before imaging.

For genome-wide screens, micrographs were obtained on the Opera (PerkinElmer) automated spinning disc confocal microscope. Three fields with Z-stacks of 5 optical sections with 0.8-μm spacing were collected per well, with each field of view containing 50–150 cells. Secondary screens were imaged on an Opera Phenix (PerkinElmer) automated microscope. All imaging was done with a 60× water-immersion objective. Acquisition settings included using a 405/488/561/640 nm primary dichroic mirror. yEGFP was excited using a 488 nm laser and emission collected through a 520/35-nm filter. tdTomato was excited using a 561-nm laser, and emission collected through a 600/40-nm filter. Dextran Alexa Fluor 647 was excited using a 640 nm laser, and emission collected through a 690/50 nm filter.

### Monitoring the formation and progression of vacuolar class G phenotype with time-lapse fluorescence microscopy

Strains *his3Δ* (DMA1) and *sec18-1* (TSA54) from the *MAT***a** deletion and TS collections were crossed to strain Y15251. Haploid FP-tagged mutant clones were selected using the SGA method. Imaging plates were prepared as described above. Imaging was done using the Opera Phenix (PerkinElmer) automated system. Z-stacks of 5 optical sections with 0.8-um spacing were first acquired at room temperature, the temperature was then shifted to 37°C, and images were acquired at 1-h time intervals for 24 h. Maximum *z*-projections, adjustment of intensity levels to optimize phenotype visualization, and image sequences were made with ImageJ (Schneider *et al*, 2012).

### Assessing endocytic vesicle formation dynamics with live-cell imaging

Strains deleted for *YDL176W* (DMA754) or *HIS3* (DMA1; wild-type control) expressing Sla1-GFP and Sac6-tdTomato were grown to mid-log phase, immobilized on ConA-coated coverslips and sealed to standard glass slides with vacuum grease (Dow Corning). Imaging was done at room temperature using a spinning disc confocal microscope (WaveFX, Quorum Technologies) connected to a DMI 6000B fluorescence microscope (Leica Microsystems) controlled by Volocity software (PerkinElmer) and equipped with an ImagEM charge-coupled device camera (Hamamatsu C9100-13, Hamamatsu Photonics) and 100×/NA1.4 Oil HCX PL APO objective. Images were acquired continuously at a rate of 1 frame/s and analysed using ImageJ (Schneider *et al*, 2012). One hundred patches from 10 to 20 cells from two independent replicates were analysed per strain. Statistical significance was assessed with the unpaired *t*-test.

### Follow-up experiments related to the assessment of incomplete penetrance

#### Penetrance as a function of replicative age or vacuole inheritance

Strains *his3Δ* (DMA1), *rrd2Δ* (DMA4876), *rpl20bΔ* (DMA4693), *cka2Δ* (DMA4484), *vac17Δ* (DMA520), and *vac8Δ* (DMA1262) from the haploid *MAT***a** deletion collection were crossed to strain BY5841. Haploid mutants expressing the three FPs (*VPH1-GFP HTA2-mCherry* and *RPL39pr-tdTomato*) were selected using the SGA method. Cells were grown to logarithmic phase in standard conditions, washed in PBS and stained with 400 μl 0.5 μg/ml CF640R wheat germ agglutinin (WGA) conjugate (CF640R WGA; Biotium) in PBS, nutating for 20 min at room temperature in the dark. Cells were then washed 3× with PBS, placed in low fluorescence medium and transferred to a ConA treated imaging plate. Acquisition of *z*-stacks was done on the Opera Phenix (PerkinElmer) automated microscope as described above. Maximum *z*-projections, channel merging and adjustment of intensity levels to optimize subcellular signal visualization (used only for figures) were made with ImageJ (Schneider *et al*, 2012). The experiment was done in biological triplicate.

#### Effect of the UPR pathway

A *URA3*::UPRE-mCherry cassette, which encodes mCherry driven by a minimal *CYC1* promoter and four tandem unfolded protein response

elements (UPREs), was amplified using PCR from pPM47 (Merksamer *et al*, 2008) and integrated at the *URA3* locus in BY4741. Primers used (URA3pr-F and dn_mCherry-R) are listed in the Reagents and Tools Table. Plasmid pPM47 was a gift from Feroz Papa. The strain with integrated UPRE-mCherry was crossed to query strains containing *SAC6-GFP::NATMX4* or *VPH1-GFP::NATMX4* and tetrads were dissected to obtain query strains with a GFP-tagged morphology marker and UPRE-mCherry (strains BY6279 and BY6285).

A mini-array of gene-deletion strains identified as intermediate penetrance mutants for actin was chosen and crossed to BY6285. Likewise, a mini-array of vacuole mutants was crossed to BY6279. SGA was used to select haploid strains with both the marked morphology compartment and the stress reporter. Cells were grown for imaging using standard conditions and imaged in low fluorescence medium containing 5 μg/ml Dextran Alexa Fluor 647 on an Opera Phenix (PerkinElmer) automated system as described above.

### Determining Single Mutant Fitness (SMF) of the DMA-SLOW collection

In order to determine the single mutant fitness for slow-growing non-essential gene-deletion strains (DMA-SLOW collection), that were previously excluded from the global genetic interaction analysis (~400 ORFs) (Costanzo *et al*, 2016), we carried out 5 SGA screens where a WT query strain carrying a *NATMX* marker inserted at a neutral locus (Y8835) was crossed to the *KANMX*-marked DMA-SLOW collection. SGA screens were performed at 30°C. Colony size was quantified using SGATools (Wagih *et al*, 2013).

### Image analysis and object quality control
#### Image pre-processing, object segmentation and quantitative feature extraction

Acquired stacks were compressed into a maximal *z*-projection using ImageJ (Schneider *et al*, 2012). CellProfiler (Carpenter *et al*, 2006) was used for object segmentation and quantitative feature extraction. Cells were segmented from intensity-inverted Dextran Alexa Fluor 647-channel images. Cell intensity measurements of the Dextran Alexa Fluor 647 channel were collected for quality control purposes. Segmented cell boundaries were then applied to the endocytic marker channel to segment secondary objects (endocytic compartments), define tertiary objects (cytoplasm) and extract area, shape, intensity and texture measurements of the segmented endocytic compartments, cytoplasm and whole cell. Two additional features were calculated from the extracted CellProfiler features: (i) fraction of the cell occupied by the screened compartment(s) (compartment_areashape_area divided by cell_areashape_area) and (ii) compartment diameter ratio (compartment_areashape_maxferetdiameter divided by compartment_areashape_minferetdiameter). In total, we extracted quantitative information for approximately 21 million single cells and approximately 73 million individual endocytic compartments. The raw data were imported into a custom-made PostgreSQL database.

#### Cell quality control

To reduce noise in the analysis due to segmentation artefacts and ensure only high-quality objects were included in downstream analyses, a quality control filter was applied to all segmented cells. First, the quality control filter discarded low-quality cell objects based on shape, size and intensity measurements collected from the Dextran Alexa Fluor 647 signal. These low-quality objects included badly

segmented cells, ghost objects (segmented background), clumps of cells and dead cells. Second, all images with < 5 cells were excluded. Third, all wells with only 1 good site (out of the 3 acquired) were excluded. Additionally, we trained a 2-layer fully connected neural network to identify and exclude from the dataset small buds that had been segmented independently of mother cells, and bad cells missed by other quality control filters (see below for details). Across all screens, the used filters discarded 20% of cell objects, leaving approximately 16.3 million good cells for subsequent analysis. On average, 640 good cells for each strain from 2.6 biological replicates were retained for downstream analyses.

### Data processing, outlier detection and classification
#### Data pre-processing

The extracted features were standardized by computing mean and standard deviation of features from wild-type control strains (negative controls) to remove feature means and scale to unit variance separately for each imaged 384-well plate in a screen. Means and standard deviations of features from each imaged plate were analysed to identify potential batch effects on plates.

#### Selection of positive controls

Positive control mutants were selected based on phenotypes annotated in the *Saccharomyces* genome database (SGD, https://www.yeastgenome.org) and published literature (Table EV1). Only mutants for which we were able to visually confirm the published phenotype in our images were included in the positive control set. Additionally, to ensure all main phenotypes were included in our classifier, an unsupervised outlier detection approach was used to search for mutants with unpublished or poorly annotated phenotypes (see below for details). The two approaches combined gave us a set of 21 different subcellular morphologies, comprising 4 wild-type and 17 mutant phenotypes. We note that our vacuole phenotypes do not perfectly overlap with the vacuolar morphological classes that have been described previously (classes A to F) (Raymond *et al*, 1992). To avoid confusion, we adopted descriptive names for most of our vacuolar phenotype classes (Figs 1C and 2).

The lists of positive control strains associated with each mutant phenotype were subsequently used to compile the classifier training set (see below for details). Visual inspection of all micrographs from positive control mutants was used to assign each mutant to a penetrance bin (100–80%, 80–60%, 60–40%, 40–20%, 20–0%). These manual penetrance assignments were used to validate the accuracy of computational penetrance assignments obtained through classification (see below for details). Positive control mutants and their manual penetrance assignments are listed in Table EV1.

#### Unsupervised outlier detection

An unsupervised outlier detection approach was used to identify additional positive control strains (see *Selection of positive controls*). First, principal component analysis (PCA) was applied to the extracted CellProfiler features to reduce the redundancy and correlation of features in the data (Hotelling, 1933). The number of PCs was selected so that at least 80% of the variance in the complete data was explained. Next, to identify mutant strains that affect the morphology of the imaged subcellular compartment, an outlier detection method was implemented with the goal of detecting cells whose morphology differed substantially from the negative (wild-

type) controls. In the feature space, we identified cells with non-wild-type morphologies based on their distances from the negative control distribution. To quantify these distances, we implemented a one-class support vector machine (OC-SVM)-based outlier detection method (Scholkopf *et al*, 2000). We used OC-SVM implemented in Python's scikit-learn package with default hyper-parameters (radial basis function kernel, kernel hyper-parameter gamma set to $1/N$ where $N$ is the number of used PCs, and hyper-parameter nu set to 0.5 in order to define a stringent population of negative control cells). For each single cell in the complete dataset, we calculated the distances to OC-SVM decision function. Next, we applied a threshold on the calculated distance at the $20^{th}$ percentile of the negative control cells to differentiate in- and outliers.

For each mutant strain, unsupervised penetrance was defined as the percentage of outlier cells obtained from the unsupervised outlier detection approach. The statistical significance of penetrance was calculated using a hypergeometric test (identical to one-tailed Fisher's exact test) (Rivals *et al*, 2007), with negative controls as the background. For each endocytic marker, top-scoring mutant strains were visually inspected to identify mutant phenotypes and additional positive control strains (see above).

### Single-cell labelling and classification

**Single-cell labelling tool** The custom-made single-cell image viewer is a Django-based web application written in Python which allows the user to input different parameters or filters and then view the cells satisfying the set conditions. The web interface is developed using HyperText Markup Language (HTML 5), Cascading Style Sheets (CSS3) and JavaScript. Taking advantage of Django's capability to use multiple databases, the primary PostgreSQL database containing raw CellProfiler features and unique cell IDs was used in this tool to pull the information needed to display each single cell. The information needed was as follows: the image to which the cell belongs, the image's location on the server and the x- and y-coordinates of the cell. The tool allows the user to label and save a phenotype for a specific cell which would then be saved to the single-cell viewer database and used to compile the cells' features for the training set.

**Manual labelling of single cells** For each marker, and separately for the primary genome-wide and secondary medium-scale screens, single cells from positive control mutant strains as well as negative control wild-type strains were manually assigned to a mutant or wild-type phenotype class using the single-cell labelling tool. In total, 42 sets of labelled cells were compiled (2 types of screens × 21 phenotypes; i.e. 4 wild-type and 17 mutant phenotypes).

For cell quality control purposes (see *Cell quality control* above), for each Sla1 and Sac6 screen, we manually labelled approximately 320 small buds, that had been segmented independently of their mother cells ("small bud" class), and approximately 250 badly segmented cells ("bad" class). We trained a 2-layer fully connected neural network (see below for details) with these two cell quality control classes and the wild-type class to identify small buds that had been segmented independently of mother cells, and bad cells missed by other quality control filters. All cells that were assigned to the "small bud" or "bad" classes with an average prediction probability across 10 random initializations of ≥ 85% were excluded from the final set of good cells.

**Training set clean-up** To identify cells in the training set that were mislabelled, we did an initial training run with all of the labelled positive and negative control cells, as described above. For training, fivefold cross-validation on the labelled dataset was used. Each fold was split into 80% for training set (20% of training set is used for validation set) and 20% for test set during neural network training. Each fold was used for training a fully connected 2 hidden-layer neural network (2NN) 10 times with different random initializations. With this approach on fivefold cross-validation and 10 random initializations, we obtain 10 separate predictions for each of the labelled cells. Any cells that were incorrectly classified in two or more random initializations were manually inspected and cells that were originally mislabelled were removed from the final labelled set. Up to 12% of the labelled cells were removed from each training set using this approach. All subsequent training of the 2NNs was on this cleaned dataset. The final number of labelled cells in each phenotype ranged from 35 to 982, with an average of 420 cells.

**Visualization of the training set feature space with t-SNE** To assess whether the extracted CellProfiler features could be used to accurately distinguish between different phenotypes, the high-dimensional feature space for each of the single cells from the training sets was visualized using a non-linear dimensionality reduction technique —t-distributed stochastic neighbour embedding (t-SNE) (Maaten & Hinton, 2008). Python's scikit-learn package (version 0.19.0) was used for t-SNE with default hyper-parameter settings except for perplexity. The perplexity hyper-parameters chosen for the 4 markers were as follows: 50 for Sac6, 30 for Sla1, 10 for Snf7, and 50 for Vph1.

**Classification: single-cell level assignment of mutant phenotypes** To classify all cells from the final dataset (see *Cell quality control* above) into different mutant phenotypes, the training sets comprised of labelled single cells were used to train a fully connected 2 hidden-layer neural network (2NN). We trained a 2NN for each of the endocytic markers and screen types, totalling eight trained 2NNs. We opted for separate training sets for the two screen types (genome-wide and secondary screen), as this strategy gave us better classification accuracy (possibly because the two screen types were imaged using different microscopes). The 2NN was implemented using Keras (https://keras.io) with TensorFlow backend (https://tensorflow.org).

The input layer consisted of the scaled single-cell CellProfiler features and we used a soft-max output layer (Bishop, 1995). The first hidden layer had 54 units and the second hidden layer had 18 units. The hidden layers used ReLU activation functions (Hahnloser *et al*, 2000). All the hyper-parameters used in the training for the Stochastic Gradient Descent optimizer (Kiefer & Wolfowitz, 1952) are specified at https://github.com/BooneAndrewsLab/ODNN. We used the same architectures and hyper-parameter settings for each network; these hyper-parameters were selected to provide good performance without overfitting, with a short training period on the whole unfiltered training sets.

For training, fivefold cross-validation on the labelled dataset was used. Each fold was split into 80% for training set (20% of training set is used for validation set) and 20% for test set during neural network training. Each fold was used for training a 2NN 10 times

with different random initializations, resulting in 10 predictions for each cell in the test set. The final probability of each cell in the test sets was calculated by averaging the probabilities from these 10 randomly initialized 2NNs. The class with the maximum average probability was used for the predicted label. The combined test set predictions were displayed in a confusion matrix and used to assess the neural network's performance. Most phenotypes could be classified with very high accuracy, except those with the smallest training sets (Fig EV2A and B).

The 2NN employs a relatively new strategy for creating an ensemble classifier; to ensure that this strategy did not create bias in its classifications, we compared it to two more traditional approaches to creating an ensemble classifier. Specifically, 10 base classifiers in the ensemble differed only in their random initializations but shared hyper-parameters and training sets. This strategy permits us to use the entire training set to train each classifier rather than only the ~68% used in bagging. For classifiers with multiple local optima, such as neural networks, the new strategy has shown better performance generalization and uncertainty calibration than bagging (Lakshminarayanan *et al*, 2017). However, to validate this method in our context, we compared it to two other approaches using the Sac6 and Vph1 genome-wide screens. One approach (2NN – CVx1) employed the same training strategy as 2NN, but predictions of unlabelled cells in the full dataset were done during each of the 10 random initializations and fivefold cross-validations. In other words, we averaged the output of 50 networks trained on five different, partially overlapping, training sets. The second approach (2NN – CVx10) did not include 10 random initializations, and the training was done using 10 independent runs of fivefold cross-validations. Here, we averaged the output of 50 networks trained on 50 different, partially overlapping, training sets. Similar to 2NN – CVx1 approach, with this third approach, we predicted the entire dataset of unlabelled cells during each of the 10 independent runs and fivefold cross-validations. We assigned cells in a screen the phenotype with the highest average probability across the 50 2NNs for both 2NN – CVx1 and 2NN – CVx10. The average correlation between these three approaches on penetrance values and phenotype fractions across all genes in the two genome-wide datasets was 0.95 (Table EV1). We thus concluded that our use of the same training sets and hyper-parameters in the ensemble in our 2NN classifier did not introduce biases compared to a scheme which employed different training sets on each classifier.

After estimating the general performance of the 2NNs using cross-validation, to make predictions on the entire dataset of unlabelled cells, we retrained the networks on the entire filtered training set. Similar to the approach described above, we trained 10 separate 2NNs starting from different random initializations and assigned cells in a screen the phenotype with the highest average probability across the 10 2NNs. Mean single-cell prediction probabilities are included in Table EV1. The 2NN classifier assigned the highest classification probabilities to those cells that were most similar to those in the manually labelled training sets (Fig 2), but at the same time allowed us to correctly assign cells with different severity of a particular phenotype to the same class.

Additionally, to identify any strains with phenotypes not included in our classifiers, we assigned all cells with low classification probabilities to a "None" class. Cells were assigned to the "None" class when the maximum probability was lower than $2/N$

(where $N$ = number of phenotypes). Visual inspection of strains with the highest fraction of cells assigned to the "None" class for each marker revealed no additional phenotypes (Table EV2). This approach does not exclude the possibility that cells with additional rare or non-penetrant phenotypes were incorrectly classified. For example, we did not include the class F phenotype (large central vacuole surrounded by several fragmented vacuolar structures) (Raymond *et al*, 1992) in our vacuole classifier, since none of the previously reported mutants had significant fractions of the population displaying the phenotype. The class F cells were therefore classified as wild-type, enlarged or multilobed.

All cells assigned a non-WT phenotype were defined as outliers. For each strain, the fraction of the cell population displaying each phenotype (specific phenotype fraction), as well as the penetrance (defined as: penetrance = 1 - % WT cells) were calculated. The phenotype fractions and penetrance calculated from 2NN classification were used in all subsequent analyses.

Custom scripts for data pre-processing, running the supervised 2 hidden-layer fully connected neural network for single-cell classification, and penetrance calculation are available at: https://thecellvision.org/tools.

### Penetrance reproducibility
#### Estimating penetrance reproducibility
To assess the quality of our computed penetrance values, we first compared them to visually assigned penetrance estimates and found a strong agreement for all four screened markers (average Pearson correlation coefficient (PCC) = 0.87) (Fig EV2C, Table EV1). We next assessed penetrance reproducibility by determining: (i) the difference in the calculated penetrance and (ii) the Pearson correlation coefficient (PCC) between replicates of mutant strains for each marker and screen type. Across all screens and markers, the average PCC between replicates is 0.65 (two-tailed $P = 0$). Finally, we focused on the replicate pairs with a penetrance difference > 30 and determined the prevalent causes leading to penetrance irreproducibility. Cell number had the biggest impact on penetrance reproducibility, since replicates with low cell counts represented 60% of replicate pairs with highest penetrance differences (Fig EV2D). Among low cell count replicates, 2/3 were from strains with considerable growth defects, making single mutant fitness (SMF) a good indicator of penetrance reproducibility. For the remaining 40% of replicates with a large penetrance difference, 42.6% could be attributed to temperature-sensitive strains (35.2% of large difference replicates) and technical artefacts (cross-contamination, bad segmentation, failed quality control or misclassification) (7.4% of large difference replicates), while for 57.4% of replicates, we could not identify a clear cause. The increased penetrance difference between biological replicates of TS strains could be a consequence of small differences in growth and imaging temperature between replicates.

#### Bootstrapping to determine sufficient cell count
Since different strains and plates varied greatly in the number of imaged cells, we used a bootstrapping approach to determine the standard deviation between replicates of varying cell counts and to estimate the minimal cell number required to obtain a confident penetrance calculation (Fig EV2E). Increasing numbers of cells were sampled from every screen individually, screens combined by

marker, and all screens combined. Sampling was done on two scales: first on a small scale ranging from 10 to 100 cells in increments of 10 and then on a larger scale ranging from 125 cells to 1,000 cells in increments of 25. Cell sampling was done one hundred times from populations of approximately 200,000–670,000 cells (for individual screens), and the average penetrance and standard deviation of the 100 independent samplings for each sample size were calculated and plotted (Fig EV2E). Based on the wide distribution of wild-type replicate penetrances (Fig EV2F), we chose a relative standard deviation of 0.2 (which is equal to approximately ± 4 penetrance points for a wild-type population) as our confidence threshold. The average minimum required cell number across different markers and screens was 98 cells, and this criterion was satisfied by 83.4% of imaged samples.

We next examined the potential impact of cell density effects, such as gradients, on penetrance, and observed no significant effects (Fig EV2G). $R^2$ between cell number and calculated penetrance for all replicates that met the minimum cell count was 0.0047.

### Identification of morphology mutants, calculation of accuracy, and false-positive and false-negative rates

#### Specific phenotype mutants (SPMs)

For each phenotype (17 mutant phenotypes representing 4 for patch actin module, 4 for patch coat module, 3 for late endosome, 6 for vacuole) and screening condition (room temperature, 30°C or 37°C), the threshold (thr) for the specific phenotype fraction was defined as the phenotype fraction value corresponding to the 98[th] percentile of the distribution of the specific phenotype fraction across all wild-type replicates. Since each 384-well imaging plate had wild-type strains at 76 positions (all border wells), a full genome-wide screen had more than 1,800 wild-type replicates. In cases where this calculated threshold was < 0.05, the threshold was set to 0.05. Additionally, the stringent threshold for the fraction of a specific phenotype was defined as: str = (max − thr) × 0.25 + thr, where max is the highest observed penetrance for that phenotype.

The final specific phenotype fraction for each mutant strain was calculated from the genome-wide and secondary screen values as the replicate number-weighted mean phenotype fraction. Each mutant strain had to satisfy the following criteria in order to be considered an SPM: (i) weighted mean phenotype fraction ≥ phenotype fraction threshold (or stringent threshold for stringent SPMs); (ii) ≥ 50 good cells and; (iii) ≥ 10 cells assigned with the phenotype in question. SPMs, stringent SPMs and thresholds are listed in Table EV2.

#### Penetrance mutants

For each marker (Sac6, Sla1, Snf7 and Vph1) and screening condition (room temperature, 30°C or 37°C), the penetrance threshold was defined as the penetrance value corresponding to the 95[th] percentile of the penetrance distribution of all wild-type replicates. The final penetrance for each mutant strain was calculated from the genome-wide and secondary screen values as the replicate number-weighted mean penetrance. Each mutant strain had to satisfy the following criteria in order to be considered a penetrance mutant for a given marker: (i) weighted mean penetrance ≥ calculated penetrance threshold; (ii) ≥ 50 good cells; and (iii) Bonferroni-corrected $P < 0.05$ (for strains not included in the secondary array). Penetrance mutants and thresholds are listed in Table EV2.

The specific phenotype and penetrance mutant groups comprise 1,623 yeast genes, of which 66.5% (1,079) were both SPMs and penetrance mutants, 25.1% were only SPMs (407), and 8.4% (137) were only penetrance mutants (also referred to as non-phenotype-specific mutants; see Fig 1D). In general, the ORFs that qualified as only non-phenotype-specific mutants or only SPMs were either (i) SPMs with significant but smaller mutant phenotype fractions that did not qualify as penetrance mutants, or (ii) non-phenotype-specific mutants with one or more mutant phenotypes with fractions below the specific phenotype significance thresholds. For example, a deletion mutant of *RRD2*, which encodes a component of a serine/threonine protein phosphatase involved in Tor1/2 signalling, had 42% of cells with aberrant vacuolar morphology, which is above the ~32% penetrance threshold for the vacuolar marker. However, none of the six specific vacuolar phenotype fractions exceeded the respective SPM thresholds (see Table EV2 for details).

We note that although a number of TS strains displayed increased levels of non-wild-type-looking cells even at room temperatures (data available at thecellvision.org/endocytosis), consistent with previous work (Li *et al*, 2011), we used only data from TS strains grown at 37°C for the identification of morphology mutants and downstream analyses.

#### Accuracy, FPR, FNR

The accuracy, false-positive (FPR) and false-negative (FNR) rates were calculated from biological replicates (same query strain, same screening condition, same microscopy setup) of mutant strains as follows:

$$Accuracy = (TP + TN)/(TP + TN + FP + FN)$$
$$FPR = FP/(FP + TN)$$
$$FNR = FN/(FN + TP)$$

A replicate pair was called a true positive (TP), if both measurements satisfied the criteria for penetrance mutants described above (penetrance > threshold, ≥ 50 cells and corrected $P < 0.05$). Similarly, a replicate pair was called a true negative (TN) when neither of the two replicates satisfied the criteria for penetrance mutants. False positives (FP) and false negatives (FN) were those pairs where one replicate was a penetrance mutant, while the other was not. The estimated average accuracy was 86.6%. The estimated false-positive rate was 9.5%. We estimate that the false positives are mainly found in the intermediate penetrance range. The false-negative rate was higher at ~24.9%, as expected from a stringent cut-off.

#### List of consensus morphology mutants

A consensus rule for genes with multiple screened alleles was used. For each marker and phenotype, a gene was considered a penetrance mutant or SPM if half or more of its alleles satisfied the respective significance criteria. Consensus morphology mutants are listed in Table EV2 (labelled as consensus_*).

For penetrance bin-dependent analysis, for genes with multiple alleles and with a penetrance mutant in the consensus list, we defined the penetrance as the maximum penetrance among the screened alleles that qualified as a penetrance mutant (see above). Penetrance bins of all consensus morphology mutants are listed in Table EV2.

### Enrichment and correlation analyses
### Data standards used in the analyses

**Protein complex standard** The protein complex standard was downloaded from the EMBL-EBI Complex Portal (https://www.ebi.ac.uk/complexportal/home) and is included in Table EV4.

**Gene ontology biological process standard** Biological process categories for functional enrichment were derived from a standard set of GO Slim biological process term sets downloaded from the Saccharomyces Genome Database (www.yeastgenome.org/).

**Biological pathway standard** The pathway standard was downloaded from the KEGG database (Kanehisa *et al*, 2014) and is included in Table EV4.

### Gene features used in the analyses: numeric features

**Marker abundance data** For each mutant strain, the mean FP intensity (CellProfiler feature cell_intensity_meanintensity) extracted from the genome-wide screens was used to calculate the relative marker abundance ("Marker relative abundance") and relative standard deviation of marker abundance ("Marker abundance CV"). For each mutant strain, the calculations were normalized to the per-plate wild-type strain values. The relative marker abundance and marker abundance CV data are included in Table EV4.

**Cell size data** For each mutant strain, raw single-cell size data (corresponding to cell area in pixels; CellProfiler feature cell_areashape_area) extracted from the genome-wide screens was used to calculate the relative mean cell size ("Relative cell size") and relative standard deviation of cell size ("Cell size CV"). For each mutant strain, the calculations were normalized to the per-plate wild-type strain values. The relative cell size and cell size CV data are included in Table EV4.

**Single mutant fitness** Single mutant fitness (SMF) values for non-essential gene-deletion strains (DMA), and essential gene temperature-sensitive strains (TSA) were taken from Costanzo *et al* (2016). For the slow-growing non-essential gene-deletion collection (DMA-SLOW), mean colony size measurements from 5 wild-type SGA screens were used to estimate single mutant fitness. Colony size was quantified using SGATools (Wagih *et al*, 2013). SMF was calculated as the relative colony size compared to wild type (Baryshnikova *et al*, 2010). The SMFs of all strains used in this study are listed in Table EV2.

**Broad conservation** Broad conservation is a count of how many species, out of a set of 86 non-yeast species, have an ortholog of a given gene. Broad conservation was assessed as described in Costanzo *et al*, 2016; (Costanzo *et al*, 2016).

**Positive, negative and total number of genetic interactions** The numbers of positive, negative and all genetic interactions ("Positive GI/Negative GI/Total GI") for each mutant strain were extracted from TheCellMap (www.thecellmap.org) (Usaj *et al*, 2017). For genes with multiple alleles, the number of GIs was averaged across alleles.

**PPI degree** Protein–protein interaction data were retrieved from BioGRID (Stark *et al*, 2011) and refer to the union of five high-throughput studies (Gavin *et al*, 2006; Krogan *et al*, 2006; Tarassov *et al*, 2008; Yu *et al*, 2008; Babu *et al*, 2012).

**Pleiotropy** Pleiotropy data were from Dudley *et al* (2005). The number of conditions (out of 22 tested) that lead to reduced fitness was used as a measure of pleiotropy.

**Multifunctionality** The total number of annotations across a set of functionally distinct GO terms described in Myers *et al* (2006) was used as a multifunctionality index. Multifunctionality was assessed as described in Costanzo *et al* (2016).

**Phenotypic capacitance** The phenotypic capacitance was used directly from Levy and Siegal (2008) and captures variability across a range of morphological phenotypes upon deletion of each of the non-essential genes.

**Co-expression degree** This measure is derived from a co-expression network based on integration of a large collection of expression datasets (Huttenhower *et al*, 2006). Co-expression degree was assessed as described in Costanzo *et al* (2016). Pairs of genes with a MEFIT value above 1.0 were defined as co-expressed.

**Expression level and transcript counts** The expression level values reflect the mRNA transcript levels of all yeast genes in wild-type cells grown in YPD measured using DNA microarrays (Holstege *et al*, 1998). Transcript counts indicate the number of mRNA copies of each transcript per cell (Lipson *et al*, 2009).

**Molecules per cell** Protein abundance data were derived from the unified protein abundance dataset compiled from 21 quantitative analyses (Ho *et al*, 2018). The "mean molecules per cell" values were used for analysis.

**Expression variance measured under different environmental conditions** For each gene, the variance in expression across all conditions surveyed in Gasch *et al* (2000) was measured. This dataset contains yeast gene expression levels measured in response to a number of different environmental conditions. For details, refer to Costanzo *et al* (2016).

**Protein abundance and localization variation** Data on protein abundance variation ("Protein abundance RV") and subcellular spread ("Subcellular spread RV") were from Handfield *et al* (2015).

**UPRE level** Data for UPR levels was from Jonikas *et al* (2009).

### Gene features used in the analyses: binary features

**Whole-genome duplicates** This binary feature reflects whether each gene has a paralog that resulted from the whole-genome duplication event (Byrne & Wolfe, 2005).

## Other datasets

**Endocytic internalization dataset** Data on endocytic internalization levels in non-essential gene-deletion mutants were from Burston *et al* (2009). Deletion strains with an invertase score below the median (no assigned value in the published dataset) were assigned a value of 0.

**Orthologs** A set of *Homo sapiens* orthologs of *S. cerevisiae* were obtained from the InParanoid eukaryotic ortholog database version 8.0 (http://inparanoid.sbc.su.se).

**Essential and non-essential gene sets** The essential and non-essential gene lists were obtained from Saccharomyces Genome Database (SGD; www.yeastgenome.org/).

## Morphology mutant enrichment analyses

**GO slim biological process** We performed the GO biological process enrichment analysis for each set of SPMs using the GO Slim mapping file available through the Saccharomyces Genome Database (www.yeastgenome.org/). SciPy's hypergeometric discrete distribution package was used to calculate *P*-values. *P*-values were adjusted using the Bonferroni correction. Fold enrichment was calculated as (mutants in term/all mutants)/(term size/all background).

**Protein complex and biological pathway** For each phenotype, we calculated the number of mutants that coded for members of each protein complex and tested for enrichment by using one-sided Fisher's exact tests. To identify specific enrichments associated with phenotypes, and not associations caused by genes that were morphology mutants in many phenotypes, we randomized the phenotype–gene associations. Then, for each randomized network, we calculated the number of morphology mutants that belonged to each complex. We only reported phenotype/complex enrichments with a Fisher's *P*-value (see "*P*_greater") below 0.05 and with phenotype/complex overlaps in the real network (see "P1") higher than 95% of the overlaps observed in the randomized networks (see "*P*_rnd"). The same approach was used to evaluate morphology mutant enrichment of KEGG biological pathways. Used standards are included in Table EV4. Enrichment results are included in Table EV3.

**Gene feature** We compared the values of morphology mutants and genes not identified as morphology mutants against a panel of gene features. We computed statistics by performing one-sided Mann–Whitney *U*-tests for numeric features ($P < 0.05$) and by one-sided Fisher's exact tests for binary features ($P < 0.05$). For features with data for multiple alleles, values for different alleles were averaged. For each numeric feature, we performed a *z*-score normalization in which we used the median (instead of the mean) and standard deviation of non-morphology mutants. Since all median *z*-scores of the non-morphology-mutant sets were centred to zero, in plots we reported only the median *z*-score values for morphology mutants. For each binary feature, we calculated the fraction of morphology mutants (f_hits) and non-morphology mutants (f_nonhits) with that particular feature. Then, we calculated the fold enrichment as the logarithm of f_hits divided by f_nonhits.

We followed the same approach to compare (i) morphology mutants and non-morphology mutants of each individual marker (vacuole, late endosome, coat and actin); (ii) penetrance mutants with high (≥ 75%), intermediate (75% > × ≥ 50%) and low (< 50%) penetrance values versus non-morphology mutants for each individual marker; and iv) SPMs versus genes that were not morphology mutants for each of the 17 mutant phenotypes. Results of these analyses are provided in Tables EV3 and EV8.

46 ORFs that are present in the screened array, but have been deleted from SGD, were excluded from analysis (list of excluded ORFs is available at thecellvision.org/endocytosis).

## Comparison of direct and indirect protein contacts

We used Interactome3D (version 2019_01) (Mosca *et al*, 2013) to select available protein complex structures in the PDB with three or more yeast proteins and identified which of the proteins in the complex were in direct contact. Interactome3D defines direct contacts between two proteins if they have at least five residue–residue contacts, which can include disulphide bridges (i.e. two sulphur atoms of a pair of cysteines at a distance ≤ 2.56 Å), hydrogen bonds (i.e. atom pairs N-O and O-N at a distance ≤ 3.5 Å), salt bridges (i.e. atom pairs N-O and O-N at a distance ≤ 5.5 Å) and van der Waals interactions (i.e. pairs of carbon atoms at a distance ≤ 5.0 Å). We classified proteins in the same complex structure that did not meet our criteria for direct contact as indirect contacts. Additionally, we compiled a list of protein pairs belonging to different protein complex structures.

For each screened gene, we built a profile with its 17 specific phenotype fractions (phenotype profile; phenotype fraction data are provided in Table EV2). For genes with several screened alleles, we used the mean specific phenotype fraction across alleles. Strains with incomplete profiles (missing data for any of the 4 markers) were excluded from the analysis. For each pair of profiles, we calculated their Pearson's correlation. Correlation values were then grouped by the relationship of proteins in experimental structures: (i) protein pairs in contact in a protein complex structure; (ii) protein pairs in the same experimentally solved protein complex structure but not in direct contact; or (iii) protein pairs that do not belong to the same solved protein complex structure. Difference between the sets of correlation values was evaluated by one-sided Mann–Whitney *U*-tests. Files with gene-averaged specific phenotype profiles, and profile PCCs are available at thecellvision.org/endocytosis.

## Functional similarity of penetrance mutants and SPMs versus non-morphology mutants

We calculated Pearson's correlation for every pair of phenotype profiles as described above. Next, we grouped gene pairs by different functional criteria: (i) gene pairs that encoded members of the same protein complex or members of different protein complexes; (ii) gene pairs that encoded proteins in the same pathway or in different pathways; (iii) gene pairs that had significantly correlated genetic interaction profiles or not (PCC > 0.2, GI PCC dataset downloaded from thecellmap.org) (Costanzo *et al*, 2016; Usaj *et al*, 2017); and (iv) gene pairs that were co-expressed or not. Functionally related gene pairs were defined as those that belong to the same protein complex or pathway, have a significant GI profile similarity, or are co-expressed. We used one-sided

Mann–Whitney *U*-tests to evaluate whether differences between the correlation sets were significant.

### Mean specific phenotype fraction per protein complex and within-complex PCC

For each protein complex and mutant phenotype, we calculated the mean specific phenotype fraction and standard deviation across genes encoding members of the complex. For genes with more than one allele screened, we used the mean phenotype fraction across alleles. We calculated Pearson's correlation for every pair of mutant phenotype profiles for genes coding for members of the complex, and calculated the mean PCC of all complex gene pairs. Results are included in Table EV7.

### Assessing mutant phenotype relatedness

**Common SPMs between mutant phenotypes** For each pair of phenotypes, we evaluated whether they tended to share more stringent SPMs than expected by chance. We calculated *P*-values using one-sided Fisher's exact tests and the false discovery rate (FDR) to correct for multiple tests.

**Phenotype similarity** For every phenotype, we built a profile using specific phenotype fraction values of all SPMs (matrix of 17 mutant phenotypes by 1,486 genes). Next, we computed Pearson's correlation coefficients across all pairs of phenotype profiles. Results are included in Table EV5. Hierarchical clustering was done using a correlation distance measure and average linkage.

**Enrichment of protein complexes between mutant phenotypes** For each pair of phenotypes, we retrieved the set of SPMs that shared both phenotypes and calculated if the common set was enriched for protein complexes. Protein complexes with *P* < 0.01 and at least 2 shared protein complex components were considered significant. Additionally, we required the overlap of common SPMs with members of a complex to be higher than 95% of the overlaps obtained in randomized phenotype–gene networks. Results are included in Table EV5.

### Quantification of follow-up experiments related to the assessment of incomplete penetrance

#### Quantifying penetrance as a function of replicative age

CellProfiler (Carpenter *et al*, 2006) was used for cell segmentation and quantitative feature extraction (including cell size and mean WGA intensity). A 2NN was used to assign each cell a phenotype class and WGA intensity was used as a proxy for replicative age. For each mutant strain, cells were sorted based on their mean WGA intensity and grouped into 6 bins as follows: 50% of cells with the lowest mean WGA intensity corresponding approximately to virgin daughters; the next 25% of cells corresponding approximately to mother cells that had undergone 1 division; the next 12.5% of cells corresponding approximately to mother cells that had undergone 2 divisions; and so on, up to the last bin containing 3.13% of the cell population with the highest WGA intensities that were assumed to have undergone 5 or more cell divisions. For each strain and ageing bin, we then determined the fraction of outliers. On average, approximately 3,800 cells were analysed for each strain for each replicate.

#### Effect of the UPR pathway: clustering of penetrance profiles

CellProfiler (Carpenter *et al*, 2006) was used for cell segmentation and quantitative feature extraction (including mean UPRE-mCherry intensity). OC-SVM outlier detection was used to assign each cell to the wild-type or outlier group. For each mutant strain, cells were sorted based on their stress response level (mean UPRE-mCherry signal intensity) and grouped into 10 bins of equal cell numbers. For each bin, we determined the fraction of outliers (unsupervised penetrance). STEM software (Ernst & Bar-Joseph, 2006) was used to cluster mutant strains into groups with distinct UPR profiles using k-means clustering. On average, 600 cells were analysed for each strain.

## Data availability

### Data

All penetrance and phenotype results are available at: https://thecellvision.org/endocytosis.

Normalized feature data of single cells used for neural network training and additional files that support the analyses are available at: https://thecellvision.org/endocytosis/supplemental.

Raw extracted quantitative features of segmented single cells used in the analysis were deposited to the Image Data Resource (https://idr.openmicroscopy.org) under accession number idr0078.

### Images

All images are available for browsing at: https://thecellvision.org/endocytosis. Raw and processed images were deposited to the Image Data Resource (https://idr.openmicroscopy.org) under accession number idr0078.

### Source code

Code for the single-cell labelling tool, unsupervised ocSVM for outlier detection, and 2 hidden-layer fully connected neural network for single-cell classification is available at: https://thecellvision.org/tools and has been deposited on GitHub:

- ODNN (https://github.com/BooneAndrewsLab/ODNN.git): scripts for data pre-processing, running supervised two hidden-layer fully connected neural network for single-cell classification, and penetrance calculation.
- One-Class SVM (https://github.com/BooneAndrewsLab/ocSVM.git): Outlier Detection with One-Class SVM.
- Single Cell Labeling Tool (https://github.com/BooneAndrewsLab/singlecelltool): custom-made graphical user interface (GUI) application that allows users to view and label single-cell images in a grid layout. Users can save a phenotype for each cell and then export the data.

**Expanded View** for this article is available online.

## Acknowledgements

We thank Michael Costanzo, Jing Hou, Sheena C. Li, Bryan-Joseph San Luis, Jolanda van Leeuwen and Chad Myers for technical help and critical comments. This work was primarily supported by grants from the Canadian Institutes for

Health Research (FDN-143264 and FDN-143265 to BA and CB, respectively) and the National Institutes of Health (R01HG00583). CB holds a Canada Research Chair (Tier 1) in Proteomics, Bioinformatics & Functional Genomics. PA acknowledges financial support from the Spanish Ministerio de Economía y Competitividad (BIO2016-77038-R) and the European Research Council (SysPharmAD: 614944). CP is supported by a Ramon y Cajal fellowship (RYC-2017-22959). QM holds a Canada CIFAR AI chair at the Vector Institute and is supported by an Ontario Institute for Cancer Research Investigator award.

## Author contributions

MMU, HF and ES carried out and analysed experiments. MMU, NS and CP performed large-scale data analysis and interpretation. MMU, NS, MU, CP, PA and AS performed additional data analysis. NS, MU and MPM developed the computer code, software and databases. MMU, NS, HF and CP created figures. MMU, HF, NS, QM, CB and BJA wrote the manuscript. QM, CB and BJA supervised the project. MMU, CB and BJA designed the project.

## Conflict of interest

The authors declare that they have no conflict of interest.

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
