## [Review Process File · Molecular Systems Biology]

Systematic genetics and single-cell imaging reveal widespread morphological pleiotropy and cell-to-cell variability

Mojca Mattiazzi Usaj, Nil Sahin, Helena Friesen, Carles Pons, Matej Usaj, Myra Paz Masinas, Ermira Shuteriqi, Aleksei Shkurin, Patrick Aloy, Quaid Morris, Charles Boone, and Brenda J. Andrews.

Review timeline:	Submission date:	18 th September 2019
	Editorial Decision:	12 th November 2019
	Revision received:	16 th December 2019
	Accepted:	15 th January 2020

Editor: Maria Polychronidou

Transaction Report:

1st Editorial Decision

12th November 2019

Thank you again for submitting your work to Molecular Systems Biology. We have now heard back from two of the three referees who agreed to evaluate your study. Unfortunately, after a series of reminders, we did not manage to obtain a report from reviewer #3. In the interest of time and since the evaluations of the two reviewers are rather similar we have decided to proceed with these two available reports. As you will see below, the reviewers think that the study is interesting. They raise however a series of concerns, which we would ask you to address in a revision.

The reviewers' recommendations are quite clear and I think that there is no need to repeat the points listed below. Please feel free to contact me in case you would like to discuss in further detail any of the issues raised by the reviewers.

REFeree REPORTS

Reviewer #1:

Summary

In the study by Usaj et al., the authors analyze the genotype-to-phenotype relationship in yeast mutants using high content imaging, automated feature extraction and neural networks-based classification at single cell resolution. The authors use the pleiotropic vesical trafficking phenotypes to describe factors contributing to phenotypic penetrance and severity. They also address the interplay between membrane trafficking and other factors including age, organelle inheritance and stress response. In general, the manuscript addresses an important aspect of high content screens such as the partial penetrance of the phenotypes, it includes very solid experiments and it is well written. We believe that for publication some points should be addressed.

Major:

- The partial penetrance is a key point of the manuscript and would be interesting to understand a little better where this heterogeneity comes from and if the single cells have a memory of their state. For example, the authors could take a not responding sub- population of cells or a subpopulation of cells with a certain phenotype and make them grow again to see if the new created population is resembling the initial sub-population or recreate a partially penetrant phenotypic population.

- Could the authors comment more extensively on the inference of the 21 used phenotypic classes? In Figure 2 not all classes are clearly distinguishable based on visual inspection (for example cortical patches actin: depolarized vs bright). Could the authors show a graphical representation of features or PC components which show clear distinction between the classes?

Along the same line of thought, the TSNE plots provided in figure S1E are not very convincing in particular for late endosome phenotypes where clear subpopulations seem to exist within the defined classes.

Maybe showing a different distance metric for mean values per phenotypic class would give a clearer positioning in feature space. Page 6 line 9, seems to be overstating the differences described in the figure S1A.

- To support the statement made on page 9 line 2-3 could the authors do GSEA to prove their point?
- Page 11 line 10-18: The proof for the functional involvement is correlative. Could the authors show functional validation of the involvement of the two genes in Golgi vesical trafficking?
- Page 13 line 4-30 and Figure 6A: Could the authors show the same analysis normalized by the wild type conditions to show that penetrance increases significantly and the observed effect is not a stochastic increase in more aberrant phenotypes with age.

Minor:

- The introduction could be a little more extensive in citing the previous work that has done phenotypic profiling of endocytosis and other screens that use the single cell resolution to analyze their screen (page 4 Line 13 and page 7 line 8-9 and not only in the discussion). For example, image based phenotypic profiling of endocytic machinery at single cell resolution also has been done by Liberali et al., 2014 where they model single cell behaviors to correct for partial penetrance behavior induced at the population level.

- Page 5 line 21: Could the authors please provide a table of the used features?

- Page 5 line 28: Could the authors please specify the threshold value used to define 'many outlier cells'?

- Figure S1H: Could the authors specify in the figure legend which wells are excluded from the penetrance calculation? (if they are excluded, otherwise would be great to have an explanation for the strong correlation between low cell number and penetrance).

- Class E is not very clear

Reviewer #2:

Review of " Systematic genetics and single cell image analysis reveals widespread pleiotropy and cell-to-cell variability" by Usaj et al.

This is a rather complex work, beyond the typically complex papers by the Andrews and Boone groups. Here, the authors push the limits of their impressive high throughput genetic techniques, to explore single cell phenotypes. They choose the endocytic pathway (from the membrane to the vacuole) to ask what genes affect it, by looking at the morphology of the structures in 4 key points in this pathway (coat, actin patch, late endosomes and vacuole), considering the whole deletion collection (not only the viable deletion collection, but including the ts-allele collection of essential genes). They label these four compartments with FPs, they find take images, they automatically find individual cells, extract morphological features, and with this information they extract 21 phenotypes for the 4 compartments, 4 WT plus 17 aberrant. To find mutants with aberrant morphology, they train a neural network to automatically classify each yeast into those phenotypes. Then, using appropriate cut-offs, determine that ~1600 mutants have "too many yeast" in one or more aberrant phenotypes. Here things get more complicated, since WT yeast also have "outliers"

with aberrant phenotypes, what brings us full-on into incomplete penetrance territory. Mutant strains have more outliers, depending on the mutant in question, the aberrant population could reach 90% of yeast (if I've not misinterpreted the results).

The paper continues, explaining their findings, into more usual SGA-type description of whole genome datasets. As in previous papers, they were able to find new functions to previously poorly characterized genes by the "guilty by association" approach, and, as usual, they get to name a few until known unnamed ORFs.

I was delighted while reading the text. The choice of the endocytic pathway is just an excuse to show the power of this approach, that can be used for other intracellular pathways/phenotypes.

My main concern is in the lack of deeper explanation of individual findings. The manuscript almost always remains at a very high level, using complex terminology and acronyms. As it is now, this paper is not going to be as useful as it could be. For the regular cell biologist it is hard to draw specific interesting findings, due to the lack of examples of each particular finding. For example, they spend a few lines explaining how mutants in ESCRT result in the morphological phenotypes they found. Well, I urge them to explain this in more detail, for ESCRT and at least for the other major complexes found: Exocyst, core-mediator, 26S proteasome, etc. And, at least try to explain what is failing in the cell that causes each of 17 aberrant phenotypes. It is obvious the authors know about this much more than they provide in text. I imagine that is due to a desire to keep the manuscript tight and focused, but the sheer cell biology/genetic impressiveness of the data deserves more explanation.

Thus, I strongly suggest the authors take the time to edit the text to include more "down to earth", cell biology findings, of which this impressive paper abounds.

Reviewer 1

In general, the manuscript addresses an important aspect of high content screens such as the partial penetrance of the phenotypes, it includes very solid experiments and it is well written. We believe that for publication some points should be addressed.

We appreciate this positive assessment of our work.

Major:

- The partial penetrance is a key point of the manuscript and would be interesting to understand a little better where this heterogeneity comes from and if the single cells have a memory of their state. For example, the authors could take a not responding sub-population of cells or a subpopulation of cells with a certain phenotype and make them grow again to see if the new created population is resembling the initial sub-population or recreate a partially penetrant phenotypic population.

The reviewer raises both an interesting question and possible line of experimentation. However, the experiments would be technically challenging with our experimental set-up (i.e. isolating the non-responding subpopulation would require either micromanipulation, or possibly the use of microfluidics if we could set things up so that cells with a given phenotype could be differentiated from those that don't have the phenotype in a population). I think it would be interesting to explore this question in the future with, perhaps, some mutants that show dramatic differences in penetrance for different phenotypes (or that represent different biological processes that cause the same phenotype). Also, we would predict that different mutant-phenotypic combinations would have different outcomes – for example, single cells that display abnormal phenotypes due to advanced replicative age, could give rise to healthy daughter cells, cell cycle-stage dependent phenotypes would only be observed in a specific subpopulation of cells, but morphology defects that are a direct consequence of the mutation may persist in a population. However, these experiments would be quite involved, and I would argue that

doing them is outside of the scope of our systematic analysis of endocytic compartment morphology.

- Could the authors comment more extensively on the inference of the 21 used phenotypic classes?

The list of 21 phenotypes was compiled based on the combination of literature search and manual inspection of highly-penetrant mutants determined by unsupervised outlier detection. We were able to find published literature for 15 of the 17 mutant phenotypes (a non-exhaustive list of papers is included in Table EV1). Details are described in the Methods sections '*Unsupervised outlier detection*' and '*Selection of positive controls*' (pages 28-29).

To clarify this step, we modified the text as follows (page 5, lines 26-30): "We used an automated unsupervised method to identify "outlier" cells with non-wild-type morphology (see Methods). To identify mutant morphologies, we visually inspected the strains with a significant fraction of outlier cells, assessed their phenotypes, and compiled a set of positive control strains, by combining published data with selected mutants (Table EV1). This approach enabled the discovery of both well-characterized and novel phenotypes...". Please see also our answer to the 3rd minor comment.

In Figure 2 not all classes are clearly distinguishable based on visual inspection (for example cortical patches actin: depolarized vs bright).

We absolutely agree with the reviewer that some of the classes are not easily distinguishable to the non-expert eye (particularly at the magnification that can be used for high throughput screens). This observation nicely shows the power of automated image analysis, where a quantitative description of images/cells/phenotypes can be used to identify features that might not be evident by visual inspection. Besides the micrographs in Figure 1C, we also included additional examples of cells as well as a cartoon for each phenotype in Figure 2 to better illustrate the different phenotypic classes that our classifiers are identifying.

From the confusion matrices (Figure EV2B), it is clear that our classifiers can distinguish between similar phenotypes. For example, the two phenotypes mentioned by the reviewer (actin patches: depolarized vs bright) are, across all screens, correctly classified in 81% and 88% of cases, with only 6-7% of cells being confused between the two classes. These phenotypes have been previously described (e.g. Moreau et al., 1996; Kaksonen et al., 2005; Table EV1) and mutants described in the literature with these morphology defects were among our true positives. Additionally, while the two phenotypes might appear similar on micrographs, they can be separated in feature space (new Figure EV1A (feature clustergram) and EV1B (actin patches t-SNE plot, classes 3 and 5)). When building the classifiers, to increase confidence in our predictions, we haven't made single-cell phenotype assignments to cells that have a maximum probability below a certain threshold (described in the Methods section, 'None' class). More information on training set sizes and classification accuracies for the genome-wide and secondary screens is also available in Table EV1.

Could the authors show a graphical representation of features or PC components which show clear distinction between the classes?

Along the same line of thought, the TSNE plots provided in figure S1E are not very convincing in particular for late endosome phenotypes where clear subpopulations seem

to exist within the defined classes. Maybe showing a different distance metric for mean values per phenotypic class would give a clearer positioning in feature space.

While the t-SNE plots might not be able to perfectly distinguish between all classes, and we see different degrees of overlap, it is important to remember that this is a 2D representation of a complex and multidimensional dataset. For example, when using PCA for dimensionality reduction (used in the unsupervised outlier detection step; described in the Methods section) the first two principal components were only able to explain between ~36 % and ~53 % of variance.

To address the reviewer's comment, we have added a panel to Figure EV1 with feature profiles for the different classes (new Figure EV1A). The added cluster maps were generated from WT-normalized feature values of all cells used in the training sets for the specified phenotypes. We hope the reviewer can appreciate, that for each pair of phenotypes, there is a cluster of features that clearly differs between the two phenotypes. We have also changed the t-SNE plot for late endosomes with one generated with a different perplexity (Figure EV1B).

Page 6 line 9, seems to be overstating the differences described in the figure S1A.

We modified the text to reflect the figure changes and address the reviewer's comment re overstating the differences:

- Main text (page 6, lines 6-10): "To confirm that the CellProfiler features derived from the cell images were sufficient to distinguish the different mutant phenotypes, we performed hierarchical clustering of the average feature values across all single cells labelled in each phenotype's training set (Fig EV1A), and non-linear dimensionality reduction using t-SNE (Maaten and Hinton, 2008) on the training set feature vectors (Fig EV1B)."

- Methods section (page 30, line 10): "..., 10 for Snf7, ..."

- EV Figure legend (page 60, lines 5-6): "(A) Hierarchical clustering of feature vectors composed of the average CellProfiler feature values across all single cells labelled for each phenotype. Average linkage and the Euclidian distance metric were used."

• To support the statement made on page 9 line 2-3 could the authors do GSEA to prove their point?

For each set of morphology mutants, we provide GO bioprocess/protein complex/KEGG pathway enrichment analysis results in Table EV3. We believe more meaningful information can be extracted using enrichment analysis approaches based on individual hit lists (Table EV3, Methods section), as opposed to GSEA on the full gene list.

To minimize any speculation, we deleted the part of the sentence pointed out by the reviewer (page 9 line 2-3 in original manuscript): "..., whereas phenotypes with few SPMs more likely indicate responses associated with a specialized pathway."

The modified sentence now reads (ending at page 9 line 3): "Phenotypes that occur in a relatively high fraction of the population in wild-type strains, such as depolarized patches or multilobed vacuoles (Fig 2), may result from a general cellular response to different stress conditions (environmental or genetic), and tend to be associated with a larger number of SPMs (Fig 2)."

• Page 11 line 10-18: The proof for the functional involvement is correlative. Could the authors show functional validation of the involvement of the two genes in Golgi vesical trafficking?

Given the scope of the study, we just propose, based on our screening and other published data, a hypothesis on the potential role of the two poorly characterized genes.

We hope experts in the Golgi-trafficking community will find the observation interesting and test our prediction.

For clarity, we revised the sentence where we are proposing a role for the two genes in vesicle trafficking (page 12, line 1): "... suggesting a possible role for these two proteins in Golgi vesicle trafficking."

- Page 13 line 4-30 and Figure 6A: Could the authors show the same analysis normalized by the wild type conditions to show that penetrance increases significantly and the observed effect is not a stochastic increase in more aberrant phenotypes with age.

We thank the reviewer for this interesting idea. We have added a new EV figure (Figure EV6) with the suggested analysis.

The manuscript has been modified as follows:

- Main text (page 14, lines 6-9): "When compared to wild-type, for four mutants (*cka2Δ*, *rpl20bΔ*, *vac8Δ*, and *vac17Δ*) we see that the relative contribution of the gene mutation decreases with each cell division (Fig EV6). This suggests that as cells get older, age-specific effects may contribute more to penetrance than gene-specific effects."

- EV Figure legends (page 64, lines 18-22): "Figure EV6 Penetrance as a function of replicative age. Related to Figure 6, Table EV8.

Bar graph showing the fraction of outliers relative to wild-type in populations of increasing replicative age (# of divisions) for 5 mutant strains (*rrd2Δ*, *rpl20bΔ*, *cka2Δ*, *vac8Δ* and *vac17Δ*). Data are presented as mean of 3 biological replicates +/- SD."

Minor:

- The introduction could be a little more extensive in citing the previous work that has done phenotypic profiling of endocytosis and other screens that use the single cell resolution to analyze their screen (page 4 Line 13 and page 7 line 8-9 and not only in the discussion). For example, image based phenotypic profiling of endocytic machinery at single cell resolution also has been done by Liberali et al., 2014 where they model single cell behaviors to correct for partial penetrance behavior induced at the population level.

We tried to keep the introduction succinct (in line with the author guidelines to only provide the necessary background information) while citing some relevant papers for both fields (high-content screening and endocytic trafficking). Due to the large body of work available for both fields, it's unfortunately impossible to reference many relevant papers. We have modified two paragraphs of the introduction to include additional references where the authors used HCS (including with single cell resolution), or screened for endocytic phenotypes:

Page 3, lines 17-25: "High-content screening, which combines HTP microscopy with multiparametric image and data analyses, provides rich phenotypic information about the spatio-temporal properties of biological systems at the single cell level (Boutros et al., 2015; Chessel and Carazo Salas, 2019; Mattiazzi Usaj et al., 2016). Large-scale screens have been productively combined with image analysis to explore different aspects of cell biology in yeast and in higher eukaryotes. For example, data on protein localization and abundance, cell shape and compartment morphology, and the prevalence of cell-to-cell variability can be quantified from cell images and the influence of genetic or environment perturbation on these cell attributes can be systematically assessed (Chong et al., 2015; de Groot et al., 2018; Heigwer et al., 2018; Styles et al., 2016; Yin et al., 2013)."

Page 4, lines 11-14: “Several large-scale studies have been conducted to identify a number of core components and regulators of the endocytic pathway in yeast and higher eukaryotes, but have largely been based on population measurements or have analysed only a subset of genes (Bonangelino et al., 2002; Burston et al., 2009; Collinet et al., 2010; Liberali et al., 2014; Seeley et al., 2002).”

- Page 5 line 21: Could the authors please provide a table of the used features?

A list of 219 CellProfiler features used in the compartment morphology analysis is now included in Table EV1. In the same table, we also include a list of 8 quality control features used for filtering badly segmented objects (described in Methods section *Cell quality control*).

- Page 5 line 28: Could the authors please specify the threshold value used to define 'many outlier cells'?

This step of the analysis was largely manual: the lists of mutant phenotypes and positive controls for each phenotype were compiled by visually inspecting images of mutant strains with high penetrance scores from the unsupervised outlier detection approach and searching published literature. Therefore, there is no 'hard-threshold' associated with the phenotype identification step. A non-exhaustive list of literature that helped us compile a list of aberrant morphological phenotypes (in combination with the unsupervised outlier detection) and positive controls for each phenotype is available in Table EV1. We have been working on automating this step, and a more automated and quantitative approach is being implemented for other projects.

Details on the unsupervised outlier detection approach used in this step are described in the Methods section '*Unsupervised outlier detection*'.

The part of the main text describing this process has been modified for clarity (page 5, lines 26-30): “We used an automated unsupervised method to identify “outlier” cells with non-wild-type morphology (see Methods). To identify mutant morphologies, we visually inspected the strains with a significant fraction of outlier cells, assessed their phenotypes, and compiled a set of positive control strains, by combining published data with selected mutants (Table EV1). This approach enabled the discovery of both well-characterized and novel phenotypes, ...”

- Figure S1H: Could the authors specify in the figure legend which wells are excluded from the penetrance calculation? (if they are excluded, otherwise would be great to have an explanation for the strong correlation between low cell number and penetrance).

We thank the reviewer for noticing this flaw. We updated the colour code; empty wells are now colored grey. We updated accordingly Figure EV2G (Figure S1H in the original manuscript) and the corresponding figure legend (page 61, lines 16-22): “(G) Evaluation of possible batch effects in the penetrance analysis. Representations of two screened plates illustrating cell count (orange) and computationally derived penetrance (blue) in each well are shown. Empty wells are coloured grey. A darker shade of orange or blue indicates increased cell number or penetrance as shown on the key below the plate representations. Even though uneven growth conditions can lead to plate-layout effects, such as gradients (top plate) or more favourable edge conditions (bottom plate), the cell density differences due to experimental artifacts do not significantly affect penetrance analysis.”

- Class E is not very clear

We are not sure what the reviewer's comment refers to exactly. Class E vacuoles are an established vacuolar phenotype, characterized by an enlarged pre-vacuolar compartment (and are often described as looking as a "diamond ring" in the literature). Examples of cells with a class E vacuole are provided in Figure 1C, Figure 2 (including a cartoon representation of the phenotype), and additional images can be found on the website (thecellvision.org/endocytosis). All mutants displaying this phenotype are listed in Table EV2 (where we capture several mutants previously known to have class E vacuoles). Additionally, a non-exhaustive list of literature used to compile a list of aberrant morphological phenotypes and positive controls for each phenotype (including literature relevant to class E) is available in Table EV1.

Reviewer #2:

This is a rather complex work, beyond the typically complex papers by the Andrews and Boone groups... I was delighted while reading the text. The choice of the endocytic pathway is just an excuse to show the power of this approach, that can be used for other intracellular pathways/phenotypes.

We are happy to hear the reviewer enjoyed reading the manuscript and appreciates the power and applicability of the described approach. The computer code for all the developed tools (unsupervised outlier detection, single-cell labeling tool, neural-network classifier) is deposited on GitHub with links provided on thecellvision.org/tools. We hope others will find the resources useful and implement them to study other pathways and phenotypes.

My main concern is in the lack of deeper explanation of individual findings. The manuscript almost always remains at a very high level, using complex terminology and acronyms. As it is now, this paper is not going to be as useful as it could be. For the regular cell biologist it is hard to draw specific interesting findings, due to the lack of examples of each particular finding. For example, they spend a few lines explaining how mutants in ESCRT result in the morphological phenotypes they found. Well, I urge them to explain this in more detail, for ESCRT and at least for the other major complexes found: Exocyst, core-mediator, 26S proteasome, etc. And, at least try to explain what is failing in the cell that causes each of 17 aberrant phenotypes. It is obvious the authors know about this much more than they provide in text. I imagine that is due to a desire to keep the manuscript tight and focused, but the sheer cell biology/genetic impressiveness of the data deserves more explanation.

Thus, I strongly suggest the authors take the time to edit the text to include more "down to earth", cell biology findings, of which this impressive paper abounds.

As the reviewer noted, due to the large amount of data and given the scope of the study, we tried to stay focused as much as possible. Describing in detail the role of each of the main protein complexes and the failings in the cell causing each of the 17 mutant phenotypes would require a substantial amount of space and 'dilute' the main message of the manuscript.

We have expanded the section of the manuscript that reports on the main protein complexes found between stringent SPMs (page 9 line 17 – page 10 line 5): "This conservative analysis identified a core set of 13 protein complexes that affect endocytic compartment morphology at multiple levels, including several protein complexes with

well characterized roles in vesicle trafficking (Fig 3B), such as the HOPS, the vacuolar t-SNARE and the retromer complexes, which are involved in anterograde and retrograde trafficking between the Golgi, endosomes and the vacuole. Mutants of these complexes have defects at the late endosome-vacuole fusion step, or defects in recycling leading to depletion of sorting machinery components, resulting in multiple 'fragmented' endosomes (kleine Balderhaar and Ungermann, 2013; Ma and Burd, 2019). Some of the related endocytic morphology defects are likely sequential, while others may stem from independent events. For example, mutations in genes encoding components of the ESCRT complexes caused three connected phenotypes: coat aggregates, condensed late endosomes, and class E vacuoles. Defects in ESCRT complex assembly and MVB formation lead to accumulation of cargo at the late endosome - all three phenotypes therefore mark an exaggerated prevacuolar endosome-like compartment (Coonrod and Stevens, 2010). In contrast, mutation of genes encoding general transcriptional regulators such as TFIID and the core mediator caused pleiotropic endocytic phenotypes which may reflect a series of independent defects in transcription.

Another core complex with effects on multiple endocytic compartments is the functionally conserved Dsl1 multisubunit tethering complex, a resident ER complex involved in retrograde Golgi-to-ER trafficking. As the upstream step in many intracellular vesicle trafficking pathways, disruption of ER-Golgi trafficking can alter both sorting through the secretory/exocytic and Golgi-to-endosome pathways, affecting both early and late endocytic compartments.”

Explaining (speculating) what is happening in the cell for each of the 17 mutant phenotypes, would require an even more substantial amount of space. 15 of the 17 phenotypes have been described in the literature (we provide a non-exhaustive list of papers in Table EV1), so we limited our discussion of phenotypes to the new vacuolar class G phenotype (page 8, lines 10-17). We did however try to include various interesting examples and short interpretations for the different phenotypes (or phenotype pairs) throughout the manuscript (for example, newly added text: page 8 lines 26-28, page 9 lines 12-13; existing: page 10 lines 20-22, page 10 line 29 - page 11 line 7, page 11 lines 19-23, page 11 lines 25-31, page 12 lines 8-15). Additionally, all GO bioprocess/protein complex/KEGG pathway enrichment analysis results are included in Table EV3, and lists of morphology mutants for each of the phenotypes in Table EV2. We hope readers interested in a particular phenotype will take advantage of these resources. Moreover, all images, penetrance and phenotype data can be browsed on the developed website (thecellvision.org/endocytosis), which also allows searching the data using multiple filters (thecellvision.org/endocytosis/advanced).

REVIEWER #3:

Overall I think the manuscript describes a rigorously performed screen, thoughtful analysis, and interesting follow-up studies. I think it is a very valuable resource for not only the yeast community, but for systems analysis in general. If it were to be published as is I wouldn't have an issue.

We thank the reviewer for this positive assessment of our work, and appreciate that the reviewer would support publication 'as is'. However, we strive to improve our work, and have included text and figure changes to address the reviewer's comments. Please see specific answers below.

I have some thoughts and suggestions:

1) This may be pedantic, but my only major issue is regarding the use of term "pleiotropy". I don't feel entirely satisfied with how the authors dealt with this. In my mind a gene is pleiotropic when it is involved in multiple distinct processes and whose mutation would result in multiple (seemingly) unrelated phenotypes. Here the authors use it if gene depletion results in multiple endocytic phenotypes. From the outset, I would not call this evidence of pleiotropy as all the phenotypes being measured are highly related to a common process (endocytosis!). So the fact that depletion of multiple genes affects multiple reporters is by itself hardly surprising. In fact the authors rightly suggest that "morphological pleiotropy" (their new definition) could be due to the fact that the reporters read out the endocytic processes at different stages that occur sequentially, or that genes (like a TF) could affect many different aspects of the same process.

We agree with the reviewer's classical definition of pleiotropy, where a gene is pleiotropic when it is involved in multiple distinct processes. To avoid confusion with this definition of pleiotropy, we introduced the term 'morphological pleiotropy' and explained its definition on page 7, lines 9-10: "We next examined the extent of morphological pleiotropy, which we define as occurring when a mutant has two or more aberrant morphological phenotypes". In the revised manuscript, we have additionally clarified this definition as described below.

As the reviewer noted, we also explain that this morphological pleiotropy could be due to sequential as well as non-sequential events. But here things get more complicated, since these non-sequential events may stem from one 'gene function' or from multiple functions (classically pleiotropic genes). Our gene feature analysis (Figure EV4B) shows that the identified morphology mutants are enriched for multifunctional genes (defined as having multiple distinct GO bioprocess annotations, Costanzo et al., 2016). Additionally, all phenotypes show some differences in GO bioprocess/protein complex/KEGG pathway enrichments (Table EV3), and the overlap of SMPs between two phenotypes is never complete (highest % of overlap is ~36%; from Table EV5).

Since imaging was done with a single marker at a time, we cannot distinguish between sequential events and co-occurring events, and our phenotype co-occurrence analysis (and extraction of core protein complexes) (Figure 3) is done solely based on stringent SPM overlap for different pairs of phenotypes. Particularly for phenotypes that are present in a small (but significant) fraction of the population, phenotype relationships thus can't be easily modelled.

Importantly, although endocytic trafficking is a process composed of sequential events (cortical patches > endosomes > vacuoles), not all of our phenotypes are directly related to this sequential process. Many of the phenotypes are reporting on a compartment morphology that is not a consequence of earlier defects in the pathway (for example class E vacuoles, class G vacuoles, the V-ATPase defect vacuoles). Additionally, several other pathways feed into endocytic trafficking, so assuming all observed phenotypes are functionally related, and all effects are largely sequential is too simplistic.

But assuming I accept the authors new definition of pleiotropy, it is unclear to me exactly how quantitatively independent/correlated each one of these endocytic phenotypes are. In fact, by looking at the co-occurrence of phenotypes, especially across reporters, they

appear to be quite well correlated. So I am not even sure ANY use of the word pleiotropy is appropriate here.

I would suggest that if the authors are going to stick to their definition of pleiotropy, they consider further clarifying the correlation/independence of classifiers/reporters/phenotypes in the presentation of the methodology and results. Specifically establishing that the classifiers unambiguously classify one phenotype from the others, and more clarification of how correlated phenotypes are (within reporter and between reporters) are across the screen would be useful. I appreciate there is some quantification here already, but I suggest the authors be crystal clear about this.

**If there is good correlation of reporters across the screen, then even the authors' use of pleiotropy I think should be re-considered.

Based on feature space representations (Figure EV1; this figure was updated in reply to reviewer#1's comment, please see above for details), and confusion matrices in Figure EV2A (additional data is included in Table EV1), different phenotypes for a specific marker can be adequately distinguished, and there is little confusion in classification of within-marker phenotypes. Each classifier was built for a specific marker (details in the Methods section 'Classification: single-cell level assignment of mutant phenotypes'), so a comparison of classification accuracy across different markers is not sensible.

To address the reviewer's concern regarding the quantification of similarity between phenotypes, we have now clustered the phenotypes based on their similarity across gene profiles composed of phenotype fractions for all genes that are SPMs for at least one phenotype (matrix of 17 mutant phenotypes x 1486 genes). These data are now presented in Figure EV4C and Table EV5.

While there is of course significant correlation between many phenotype pairs, it's evident that not all phenotypes are related. From the phenotype similarity analysis (Table EV5): 56 pairs are significantly correlated (PCC p-value < 0.05; 8 pairs are within-marker, 48 are between different markers), 30 pairs are significantly anti-correlated (10 pairs are within-marker, 20 are between different markers), and 50 pairs are not significantly correlated (12 within-marker, 38 between different markers). The phenotype pairs that are not significantly correlated share 340 unique genes (which is ~23% of all SPMs).

To sum up: To avoid misinterpretations and better present our results on 'morphological pleiotropy', we have made several changes:

- Title: Added the word 'morphological'. The new title is "Systematic genetics and single cell imaging reveals widespread morphological pleiotropy and cell-to-cell variability"
- Introduction and Discussion: Added the word 'morphological' at page 4 line 16, and page 16 line 3.
- Main text (page 7, lines 9-12): "We next examined the extent of morphological pleiotropy, which we define as occurring when a mutant has two or more aberrant morphological phenotypes. It is important to note that morphological pleiotropy does not necessarily imply functional pleiotropy, where a gene affects multiple functionally distinct processes (Paaby and Rockman, 2013)."
- Main text (page 9, lines 8-13): "To better understand the relationships between different phenotypes, we measured the pairwise correlations between each of the 17 mutant phenotypes across all SPMs (Fig EV4C, Table EV5). As expected, this comparison revealed a large number of correlated phenotype pairs; however, 86 phenotype pairs (of the 136 possible) were either not significantly correlated or were anti-correlated (Table EV5); suggesting orthogonal and opposite cellular events. For example, enlarged and

multilobed vacuoles are anti-correlated, consistent with defects in either membrane fission or fusion.”

- Figure EV4C (legend on page 63, lines 10-12): “Heatmap of pairwise Pearson correlations between the 17 mutant phenotypes. A more intense blue colour indicates a higher PCC (scale bar at the top left). Unsupervised hierarchical clustering was performed using the correlation metric and average linkage.”

- Methods (page 42, lines 18-21): “Phenotype similarity: For every phenotype we built a profile using specific phenotype fraction values of all SPMs (matrix of 17 mutant phenotypes by 1486 genes). Next, we computed Pearson’s correlation coefficients across all pairs of phenotype profiles. Results are included in Table EV5. Hierarchical clustering was done using a correlation distance measure and average linkage.”

- Table EV5: added tabs ‘mean mutant phenotype profile’, ‘mutant phenotype similarity’.

All this is to say that while I think the use of multiple reporters is powerful, sheds light on function and network position/connectivity (of this I am completely convinced), I am not sure there is evidence here of pleiotropy by any definition.

Lastly, I do wonder whether the authors missed a bit of a trick. Because the reporters do read out largely sequential process, I wonder whether the number of phenotypes provides information how the dynamics of gene function (similar to the use of the "Hierarchical Interaction Score" in Snijder et al. Nature Methods 2013)? As mentioned above, not all phenotypes are directly associated with endocytic trafficking, and would thus not be part of a series of linear events. Additionally, our data does not provide any information about whether two phenotypes from two different markers coexist at the single-cell level. Combined with the widespread incomplete penetrance, we feel interpreting many of the gene or phenotype relationships would be difficult and potentially misleading.

While HIS is a nice approach to infer hierarchical relationships, we are not sure that the approach is ideal for our data. For example, in Liberali et al 2014, endocytic activity, defined as ‘a normalized log-10 mean intensity per cell’, was used to compare RNAi effects across the 13 screens and to infer functional interactions. The HIS calculation and other systems level analyses were done based on a z-score of the mean endocytic activity for each gene and each assay (i.e. population-context normalized mean cell intensity per gene), so all screens report some measure of normalized mean intensity for each gene, and are directly comparable. Our final data matrix, on the other hand, is completely removed from the original CellProfiler features, and reports the fraction of mutant cells in a population (please see also paragraph above).

2) I am not entirely sure I understand what the model is regarding incomplete penetrance. Are the authors saying that a gene which affects replication rates, oxidative stress, asymmetric inheritance will also affect the penetrance of a mutant phenotype? So are they claiming these are the primary drivers of penetrance? Or just some possibly factors to consider?

The factors we explored with small follow-up experiments (stress response, replicative aging, organelle inheritance) are just some of the possible factors that contribute to incomplete penetrance. In the discussion section we say (page 17, line 25-28): “In fact, for the majority of mutants, variability in morphological phenotypes between individual cells in an isogenic cell population is likely not driven solely by a genotype-to-phenotype

relationship, but rather by a combination of smaller contributions from various effects that impact single cells differently depending on their physiological state.”

We modified the text for clarity:

- (page 13, lines 15-18): “Our quantitative single-cell analysis of the morphological defects associated with each marker provided a unique opportunity to explore some of the potential molecular and cellular mechanisms underlying incomplete penetrance.”

- (page 15, lines 30-31; added to the end of this section): “These experiments show that replicative age, organelle inheritance, and response to stress are among the possible factors that contribute to incomplete penetrance in isogenic populations.”

3) I think the authors should look at Yin et al., Nature Cell Biology 2013 (PMID: 23748611), where morphological heterogeneity and penetrance were examined very closely in the context of high-content screen. Moreover, Yin et al BMC Bioinformatics 2008 (PMID:18534020) was to my knowledge the first to use outlier detection to generate new classes in high-content screens.

We thank the reviewer for these literature suggestions. We found both very useful and have also included Yin et al., 2013 in the introduction, as an example where HCS was used to assess heterogeneity.

For this project, we have compiled the list of phenotypes and positive controls for each phenotype by visually inspecting images of mutant strains with high penetrance scores calculated with an unsupervised outlier detection method (described in the Methods section) and searching published literature. We have been working on automating this step, and a more automated and quantitative approach is being implemented in other projects. Yin et al., 2008 will be a valuable resource for further refining our approach to unsupervised phenotype identification.

Other changes related to editorial comments:

Please note that our editorial policy does not allow “Data not shown”. We have removed from the text the following sentence (page 6, row 12 in the original manuscript file): “We used CellProfiler features instead of those learned using a convolutional neural network (CNN) because, unlike recent studies (Durr and Sick, 2016; Eulenberg et al., 2017; Kraus et al., 2017; Parnamaa and Parts, 2017), the CNN performed poorly on our relatively small training set (data not shown).”

In line with the Author guidelines for Title length and reviewer #3 comments, we have changed the manuscript title to “Systematic genetics and single cell imaging reveals widespread morphological pleiotropy and cell-to-cell variability”.

We hope you find our revisions satisfactory, and look forward to hearing from you.

With warm regards,

Accepted

15th January 2020

Thank you again for sending us your revised manuscript. We are now satisfied with the modifications made and I am pleased to inform you that your paper has been accepted for publication.

Corresponding Author Name: Dr. Brenda J. Andrews

Manuscript Number: MSB-19-9243